# Genome-resolved year-round dynamics reveal a broad range of giant virus microdiversity

Yue Fang,[1] Lingjie Meng,[1] Jun Xia,[1] Yasuhiro Gotoh,[2] Tetsuya Hayashi,[2] Keizo Nagasaki,[3] Hisashi Endo,[1] Yusuke Okazaki,[1] Hiroyuki Ogata[1]

**ABSTRACT** Giant viruses are crucial for marine ecosystem dynamics because they regulate microeukaryotic community structure, accelerate carbon and nutrient cycles, and drive the evolution of their hosts through co-evolutionary processes. Previously reported long-term observations revealed that these viruses display seasonal fluctuations in abundance. However, the underlying genetic mechanisms driving such dynamics of these viruses remain largely unknown. In this study, we investigated the dynamics of giant viruses using time-series metagenomes from eutrophic coastal seawater samples collected over 20 months. A newly developed computational pipeline generated 1,065 high-quality genomes covering six major giant virus lineages. These genomic data revealed year-round recovery of the viral community structure at the study site and distinct dynamics of viral populations that were classified as persistent ($n = 9$), seasonal ($n = 389$), sporadic ($n = 318$), or others. By profiling the intra-species nucleotide-resolved microdiversity through read mapping, we also identified year-round recovery dynamics at subpopulation level for viruses classified as persistent or seasonal. Our results further indicated that giant viruses with broader niche breadth tended to exhibit higher levels of microdiversity. We argue that greater microdiversity of viruses likely enhances adaptability and thus survival under the virus–host arms race during prolonged interactions with their hosts.

**IMPORTANCE** Recent genome-resolved metagenomic surveys have uncovered the vast genomic diversity of giant viruses, which play significant roles in aquatic ecosystems by acting as bloom terminators and influencing biogeochemical cycles. However, the relationship between the ecological dynamics of giant viruses and underlying genetic structures of viral populations remains unresolved. In this study, we performed deep metagenomic sequencing of seawater samples collected across a time-series from a coastal area in Japan. The results revealed a significant positive correlation between microdiversity and temporal persistence of giant virus populations, suggesting that population structure is a crucial factor for adaptation and survival in the interactions with their hosts.

**KEYWORDS** giant virus, microdiversity, metagenome, *Nucleocytoviricota*

Microdiversity refers to subspecies-level (intra-population) genomic diversity (1–3). It can alter physiological characteristics (4), differentiate ecological niches (5), and maintain the stability of microbial populations (6). The driving forces that influence microdiversity include genetic factors, such as mutations, horizontal gene transfer, and genomic rearrangements, and selective pressures, such as temperature, light, predator, and nutrition (3, 7–9). Microdiversity is shaped by the complex interplay of these factors and therefore provides a framework for understanding the eco-evolutionary dynamics of the microbial world. Recent metagenomics studies started to investigate

**Peer Reviewers** Linxing Chen, University of Science and Technology of China, Hefei, Anhui, China; Frank O. Aylward, Virginia Polytechnic Institute and State University, Blacksburg, Virginia, USA

Address correspondence to Hiroyuki Ogata, ogata@kuicr.kyoto-u.ac.jp.

Yue Fang and Lingjie Meng contributed equally to this article. Author order was determined alphabetically.

The authors declare no conflict of interest.

See the funding table on p. 19.

the microdiversity of environmental viruses (mainly small bacterial viruses) (10, 11), revealing temporal changes of microdiversity and relationships between microdiversity and biogeography. Additionally, an increasing body of literature has underscored the importance of subspecies variation in viruses. A few mutations in a single protein, such as the spike protein in coronaviruses, can significantly alter viral infectivity and, consequently, epidemicity (12). Similarly, one single mutation expands the host range of *Pseudomonas* phage LUZ7 (13). Therefore, understanding virus microdiversity is crucial for enhancing our knowledge on their impact in nature.

Viruses play a critical role in marine ecosystems by modulating microbial community composition and participating in biogeochemical cycles (14, 15). Members of the viral phylum *Nucleocytoviricota*, often referred to as giant viruses, are widespread (16, 17), abundant (18), and active (19, 20) in the ocean. Some of these viruses contribute to the global carbon export by infecting plankton (21). Temporal dynamics are known to reflect the interactions of giant viruses with their hosts and the population structure under environmental pressures. *Emiliania huxleyi* viruses tend to retain the same genotypes throughout the bloom periods of their hosts (22). A coastal *Imitervirales* community exhibits synchronous seasonal cycles with eukaryotes and year-round recurrence; nonetheless, most individual viral populations tend to be specialists rather than generalists (23). Similarly, distinct patterns in the seasonality of individual giant viruses at Station ALOHA were observed (24). To determine what underlying forces are driving viral niche differentiation, a thorough and comprehensive investigation is needed into the structure, dynamics, and diversity of viruses at the subspecies level.

Profiling giant viral microdiversity and dynamics has numerous challenges. Previous studies predominantly employed marker-based approaches, which do not capture variability for all populations (25) nor offer genome-wide evidence of selection (26). Therefore, high-quality and fine-resolution genomes are necessary for comprehensive investigation. Additionally, the scarcity of time-series data limits the ability to track giant virus population dynamics over time. Moreover, achieving adequate sequencing depth is also necessary for assembling and capturing signals of microdiversity, especially because the lower abundance of giant viruses makes them harder to detect compared with the overwhelming signals of prokaryotes and phages (14, 18).

Here, we analyzed 42 coastal samples collected during 20 months from January 2017 to September 2018. The time-series samples were subjected to deep sequencing, which yielded over 1.8 Tbp metagenomic reads. We also developed a nearly automatic pipeline for generating metagenome-assembled genomes (MAGs) of giant viruses; this pipeline consisted of assembly, binning, screening, quality control, deduplication, and quality assessment. Applying this pipeline to 42 coastal metagenomes resulted in the creation of a coastal giant virus genome database containing 1,065 non-redundant giant virus MAGs that serve as representative species-level genomes. Furthermore, we profiled the intra-species nucleotide-resolved microdiversity of giant viruses through metagenomic read mapping and revealed their population structure over 20 months across 2 years. Collectively, the results from these two methods provided fine-scale insights into the ecological roles and evolutionary trajectory of giant viruses.

## RESULTS

### Pipeline for generating giant virus genomes

In this study, we designed an automated pipeline for generating MAGs of giant viruses. The pipeline incorporates the following steps. First, metagenomic reads from each sample are individually assembled. Reads from environmental samples are then cross-mapped to each contig sets (i.e., contigs from each sample are mapped with reads from all samples), and the contigs are binned based on coverage depth and nucleotide frequency (Fig. 1a). Potential giant virus bins are subsequently identified using a marker gene density index (see Methods). The delineation and refinement of the potential giant virus bins to generate giant virus MAGs involve a multi-step process using programs that detect *bona fide* viral contigs, including removing non-giant virus bins and contigs, and

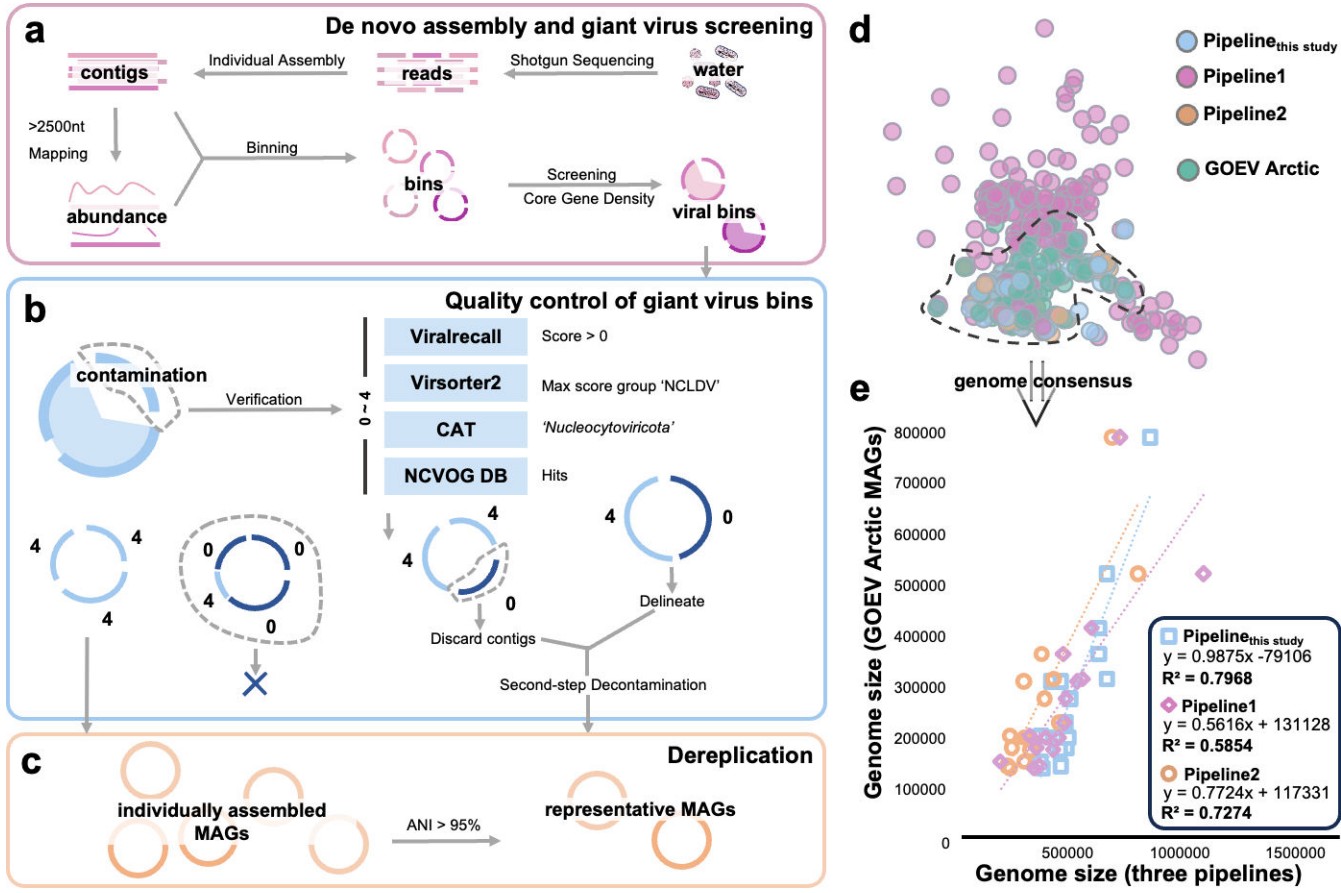

**FIG 1** Pipeline for generating high-quality MAGs of giant viruses and an assessment. Schematic flowchart illustrating the process for generating high-quality MAGs of giant viruses, including steps of (a) *de novo* assembly and giant virus binning; (b) quality control and refinement of giant viruses; (c) dereplication. (d) Gene-sharing patterns among the MAGs generated by the three different pipelines using a test 28 *Tara* Arctic metagenomes and MAGs from the same samples in the GOEV database. Refer to supplementary information for details of the benchmark analysis. Each node represents a MAG, and nodes are connected based on shared OGs. (e) Comparison of genome sizes of MAGs produced by the three pipelines with those in the GOEV database. Each dot represents a comparison of genome sizes between MAG pairs within the same genome consensus, as generated by the pipelines and recorded in the GOEV database. The x-axis represents the genome size of MAGs generated by the three pipelines, and the y-axis represents the genome size of the corresponding MAGs in the GOEV database. A best-fit linear trendline is shown.

resolving chimeric bins (Fig. 1b). The final giant virus MAG data set is then generated by deduplicating MAGs with a 95% average nucleotide identity (ANI) threshold (Fig. 1c).

Prior to applying our pipeline to our coastal metagenome samples, we conducted an evaluation of the pipeline using previously published 28 *Tara* Oceans Arctic metagenomes of protist-size fractions, for which manually curated giant virus MAG data are available. We compared our pipeline with two previously reported computational pipelines used in global giant virus metagenomic surveys. We first assessed the completeness and contamination of MAGs generated by the three pipelines (see supplementary information) and found that the pipeline developed in this study effectively controlled contamination rates while producing a sufficient number of complete MAGs. Subsequently, we aligned the MAGs with a manually curated virus genomes from the same metagenome data (the GOEV database) and demonstrated that our pipeline more accurately reflects genome size after delineation under manual supervision (with a slope close to 1; Fig. 1d and e) than the other two pipelines. Taken together, the benchmark results indicated that our pipeline provides a balanced workflow, capable of reconstructing low-contamination genomes and recovering diverse viral lineages.

## Japan coastal giant virus genomes

We then applied the pipeline (Fig. 1a through c) on 42 coastal metagenomes extracted from water samples of 0.22–0.8 µm size fractions in Uranouchi Inlet, Kochi Prefecture, Japan (Fig. S1; Table S1). A total of 2,655,994 contigs were assembled, and 16,110 raw bins were generated. Initially, we screened 3,082 potential giant virus bins (19.13% of all raw bins; range, 25–147 from individual metagenomes). Subsequently, we refined them to enhance their quality and retained 2,635 giant virus MAGs (16.4% of the total bins). The pairwise ANI of these giant virus MAGs showed a bimodal distribution akin to that observed in prokaryotes (Fig. S3) (27), which indicated that the boundary for populations of coastal giant viruses was around 95% ANI; this was consistent with findings from a previous study (28). As a result, a collection of 1,065 nonredundant giant virus MAGs was established to represent giant virus populations in Uranouchi Inlet. These genomes were then used as the species-level references (95% ANI) for microdiversity analyses.

Phylogeny-informed MAG assessment (PIMA) approach, developed in this study that uses core genes of phylogenetically closely related genomes to assess their quality (see supplementary information), was performed to assess the quality of 1,065 representative giant virus MAGs (Fig. S4a; Tables S2 and S3). A total of 68 MAGs were not evaluated by PIMA because they were classified as long branches and did not belong to any clade within our designated threshold for assessment. The quality of the remaining 997 MAGs was also estimated using CheckV (Table S3). The median consistency and redundancy values for the MAGs by PIMA were 83.33% and 14.81%, respectively (Fig. S4a). A notable proportion of MAGs ($N$ = 392) exhibited consistency greater than 80% and redundancy less than 20%. CheckV assessment helped identify 14 complete and 171 high-quality MAGs. Together, these assessment results demonstrated a strong performance of the pipeline when applied to the Uranouchi metagenomes.

## Community composition of giant virus MAGs

We conducted taxonomic classification by reconstructing a phylogenomic tree using concatenated amino acid sequences of three marker genes: RNA polymerase alpha subunit (RNAPa), RNA polymerase beta subunit (RNAPb), and DNA polymerase B (DNApolB) (Fig. 2a). Out of 1,065 giant virus MAGs, 1,052 were taxonomically categorized within known orders of *Nucleocytoviricota*, including *Imitervirales*, *Algavirales*, *Asfuvirales*, *Pimascovirales*, and *Pandoravirales*, and the remaining 13 genomes were mirusviruses. The proportions of the number of viruses in different lineages observed in Uranouchi Inlet showed a similar but slightly different trend compared with those noted in global oceanic surveys. Of the 1,052 *Nucleocytoviricota* MAGs, 831 were classified into the order *Imitervirales* (78.99%), which is a higher proportion than was detected in the *Tara* Oceans project (66.19%) (29) and indicates a greater diversity of hosts for *Imitervirales* in the coastal waters. The size of the giant virus MAGs ranged from 200 Kbp to 1.9 Mbp. We discovered three clades represented by 22 MAGs that together formed a sister clade close to the *Ectocarpus siliculosus* virus group, which includes viruses that infect brown algae (Fig. S5). One of the clades (CladeC in Fig. S5; $N$ = 12) was characterized by large genomes ranging from 1.2 to 1.9 Mbp (median = 1.7 Mbp).

Additionally, we identified 13 MAGs affiliated with the recently identified viruses in the phylum "*Mirusviricota*" (Fig. 2a) (28). The 13 MAGs encoded a nearly complete set of core genes commonly found in previously discovered mirusviruses. The largest mirusvirus MAG from Uranouchi Inlet had a genome size exceeding 484 Kbp (UUJ171113_122), which surpassed all previously discovered mirusvirus genomes. One MAG, UUJ180313_81, showed 86% ANI and wide alignment breadth with a nearly complete mirusvirus genome detected in the Mediterranean Sea (Fig. S4b). The size of this MAG (>398 Kbp) was almost as large as the nearly complete continuous mirusvirus genome (432 Kbp) (28), which demonstrated that the assembly and binning process was appropriate for detecting mirusvirus genomes. Reconstructing the phylogenomic tree of these 13 MAGs with previous mirusviruses revealed that six MAGs from Uranouchi Inlet corresponded to family M01, whereas the other seven MAGs were classified in family

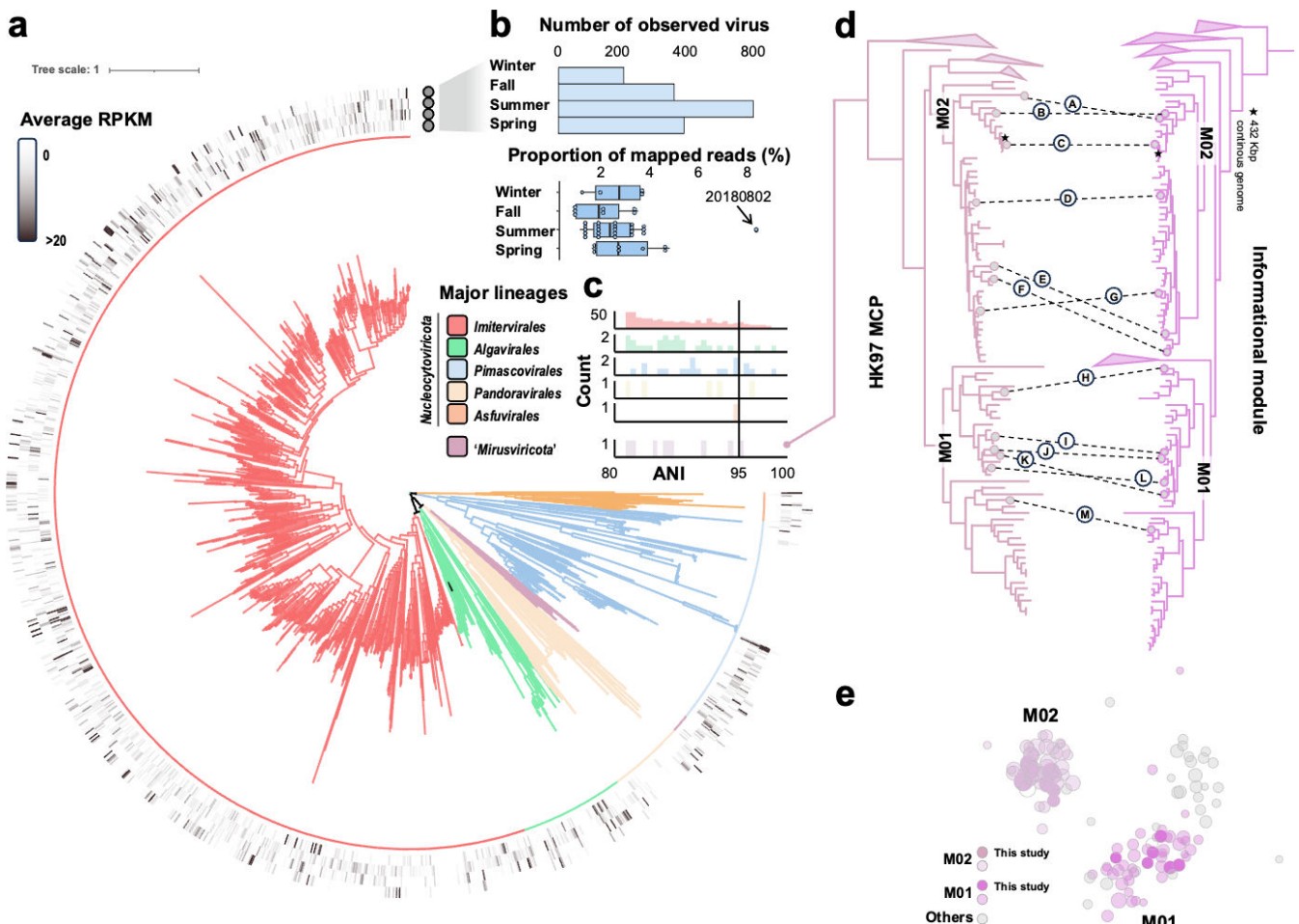

FIG 2  Community composition of giant viruses in Uranouchi Inlet. (a) Phylogenetic tree of 1,065 MAGs. The maximum likelihood tree, constructed from a concatenated alignment of three hallmark genes (DNApolB, RNApolA, and RNApolB), showcases the diversity among giant viruses. The tree is color coded to represent six lineages used in subsequent analyses: *Imitervirales*, *Algavirales*, *Pandoravirales*, "*Mirusviricota*," *Pimascovirales*, and *Asfuvirales*. (b) Number of observed giant virus MAGs and the proportion of mapped reads to giant viruses relative to all sequenced reads among four seasons. (c) Number of MAGs with high ANI to the genomes in three public global genome datasets (28, 30, 31). (d) Maximum likelihood phylogenetic tree built from the "*Mirusviricota*" MAGs based on a concatenated alignment of three hallmark informational genes (RNApolA, RNApolB, and DNApolB), right; maximum likelihood phylogenetic tree built from the "*Mirusviricota*" MAGs of the HK97-fold major capsid protein, left. Dashed lines on the two phylogenetic trees indicate the positions of 13 MAGs identified in Uranouchi Inlet. (e) Bipartite networks of "*Mirusviricota*" MAGs. The two colored groups represent two families of *Mirusviricota*, M01 and M02, and gray circles represent five other families defined in a previous study (28). The darker circles represent the MAGs assembled in this study, whereas the lighter circles are from the previous study.

M02 (Fig. 2d). M01 and M02 are the most diverse of the seven recognized families of the "*Mirusviricota*" phylum (28). This classification was also supported by the gene content of those MAGs (Fig. 2e). Additionally, the presence of genes encoding the HK97-fold major capsid protein in the Uranouchi MAGs supported their identification as mirusviruses. In the following analyses, we used five *Nucleocytoviricota* orders, *Imitervirales*, *Algavirales*, *Asfuvirales*, *Pimascovirales*, and *Pandoravirales*, and one viral phylum, "*Mirusviricota*," as the main lineages.

## Community and population dynamics of giant viruses

Giant virus community diversity (Fig. 3a) and composition at the main lineage level (Fig. 3b) showed variations during the sampling period from January 2017 to September 2018. Lineage-level diversity increased during the summer months across 2 years (Fig. 3b). Overall, *Imitervirales* were constantly abundant at any given time point

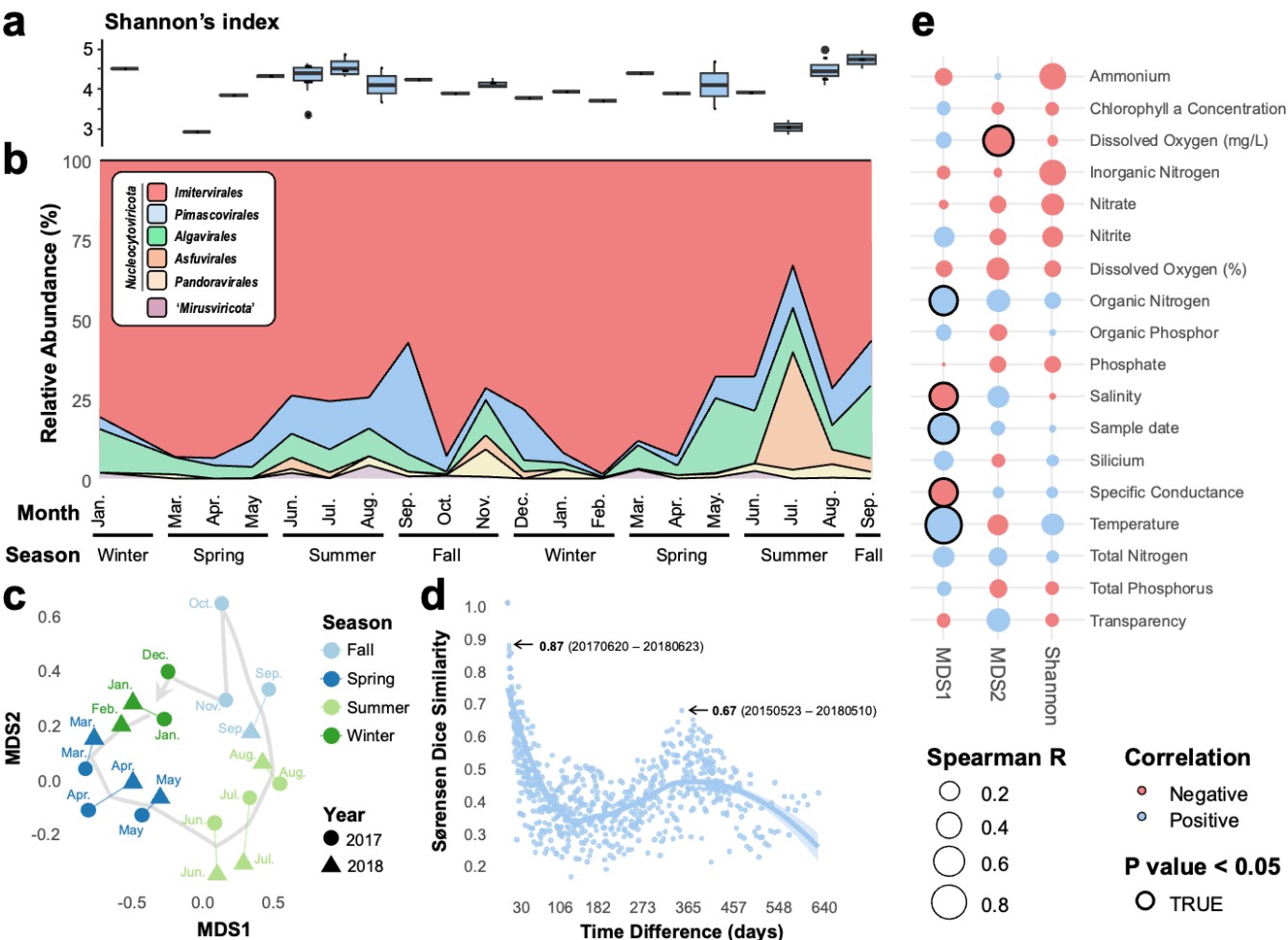

**FIG 3** Time-series variation of giant virus community in Uranouchi Inlet. (a) Shannon's diversity index for the total giant virus community. (b) The composition of the viral community at the lineage level. (c) NMDS ordination plot based on Sørensen–Dice dissimilarity between communities from January 2017 to September 2018. The stress value of the NMDS was 0.03. (d) Pairwise community similarity, which compares the similarity between viral communities from two samples, analyzed over time intervals ranging from 0 to 640 days. The community similarity level is given by Sørensen–Dice dissimilarity based on the presence/absence of species in the community. (e) Spearman's correlation coefficients between environmental factors and viral community. Correlation analyses were performed for MDS1 and MDS2 of panel (c) and Shannon's index of the viral community. $P$ values were adjusted by the Benjamini–Hochberg procedure.

and constituted the majority of the giant virus community all the time, followed by *Pimascovirales* and *Algavirales*. However, the abundance of many giant virus lineages demonstrated distinct seasonal dynamics (Fig. S6a). Specifically, *Imitervirales* displayed a relatively persistent presence throughout the 2-year period, whereas enrichment of other giant virus lineages occurred during the summer months (i.e., June, July, and August); this indicated seasonal flourishing of a diverse range of their hosts. Among the top 50 most abundant giant virus MAGs (cumulative reads per kilobase per million [RPKM] of all samples for each MAG), the majority belonged to the orders *Imitervirales*, *Pimascovirales*, and *Algavirales* (Fig. S7a). However, the most abundant MAG, *Heterocapsa circularisquama* DNA virus (HcDNAV) (32), which belongs to *Asfuvirales* and infects the toxic bloom-forming dinoflagellate *Heterocapsa circularisquama*, showed remarkable abundance in July 2018 (maximum RPKM, 992.71), which accounted for 35.2% of the total viral community (Fig. 3b; Fig. S7a).

Non-metric multidimensional scaling (NMDS) ordination of the data sets revealed a clear year-round cycle of the entire giant virus community (Fig. 3c), which demonstrated a month-to-month succession from January to December. Year-round recovery of the community was also demonstrated by a community similarity analysis (Fig. 3d).

The pairwise community similarity showed a cyclical yearly pattern with a peak at an approximately 365-day interval. Two samples from May across the 2 years exhibited a similarity index of 0.67. Temperature and dissolved oxygen were the two most significant environmental variables related to the year-round recovery of the viral community (Fig. 3e).

At the population level, the frequency of giant virus MAGs being abundant over a period of 20 months exhibited a right-skewed distribution pattern, with 90.06% of MAGs appearing in fewer than 5 months. MAGs that were present for five or more months were predominantly from *Imitervirales*, *Pimascovirales*, and *Algavirales* (Fig. S7b). Only 21 MAGs were present in four seasons (Fig. 4a; present in at least 1 month of all seasons). Distinct distribution patterns were evident in the dynamics of giant virus MAGs. Based on the population dynamic patterns, giant virus MAGs that had appeared before the second winter (December 2017) ($N = 744$) were categorized into three niche groups (persistent [$n = 9$], seasonal [$n = 389$], and sporadic [$n = 318$]) or as "other" ($n = 28$) (Fig. 4; see Methods). The proportion of MAGs that belonged to these categories are shown in Figure 4b. Among all the six lineages, *Algavirales* was the only one with the seasonal population category being dominant (68.9%). *Imitervirales* and *Pimascovirales* included persistent viral populations. *Asfuvirales* had the highest proportion of sporadic populations (75%). The typical dynamic pattern of each category is provided in Figure 4c.

## Persistence and microdiversity

Then, we investigated factors that are associated with niche breadth (persistence) measured by the Levins' index (Fig. 5a), which takes into account the number of months a taxon occupies and its relative contributions to respective months, with higher values indicating generalist taxa that are equally abundant across multiple months. The strongest correlation was observed between the Levins' index and the average nucleotide diversity (ND) (Spearman's correlation coefficient 0.41, $P$ value < 0.001) (Fig. 5a). Consistent with this observation, the average ND values of persistent and seasonal giant virus populations were higher than that of sporadic populations (Fig. S8a). Similarly, the average ND of MAGs appearing in multiple seasons tended to be higher than that of MAGs occurring in only one season (Fig. S8b). Because the viral community exhibited a year-round recovery pattern, we investigated whether the recurrence of individual MAGs was related to their microdiversity. The average ND of recurrent MAGs (those present in both 2017 and 2018) was significantly higher compared with that of non-recurrent MAGs ($P$ value < 0.001) (Fig. S8c). Overall, giant viruses with higher persistence levels displayed higher ND (Fig. 5c). At the lineage level, *Algavirales* and *Imitervirales* had the highest median ND, followed by "*Mirusviricota*," *Pimascovirales*, *Pandoravirales*, and *Asfuvirales* (Fig. 5b). All lineages exhibited higher ND in the niche categories associated with generalists (persistent and seasonal) compared with that of specialists (sporadic), although this tendency was not statistically significant for "*Mirusviricota*" and *Pandoravirales* (Fig. S9d).

The microdiversity of giant viruses, assessed through ND and SNV/Mb, was not strongly correlated with genome size ($R^2 = 0.02$ and 0.06, respectively; Fig. 5a; Fig. S9c). Additionally, to address potential biases in microdiversity detection due to sequencing depth and mapping approaches, we analyzed the relationship between microdiversity and the read coverage of the analyzed sample (i.e., the highest RPKM across samples). ND and SNV/Mb were not influenced by coverage depth ($R^2 = 0.02$ and 0.05, respectively) as much as niche breadth (Fig. S9a and b). Both results indicate that our microdiversity measurement was not significantly influenced by possible artifacts in binning and mapping.

Finally, we explored the temporal dynamics of microdiversity within viral populations. First, we calculated the fixation index of each viral population (i.e., MAG) across 42 samples in 20 months to assess the pairwise distance of the genetic structure of the viral population among samples. Similar to the community level analysis, overall year-round recovery patterns of microdiversity were observed in both the persistent and seasonal

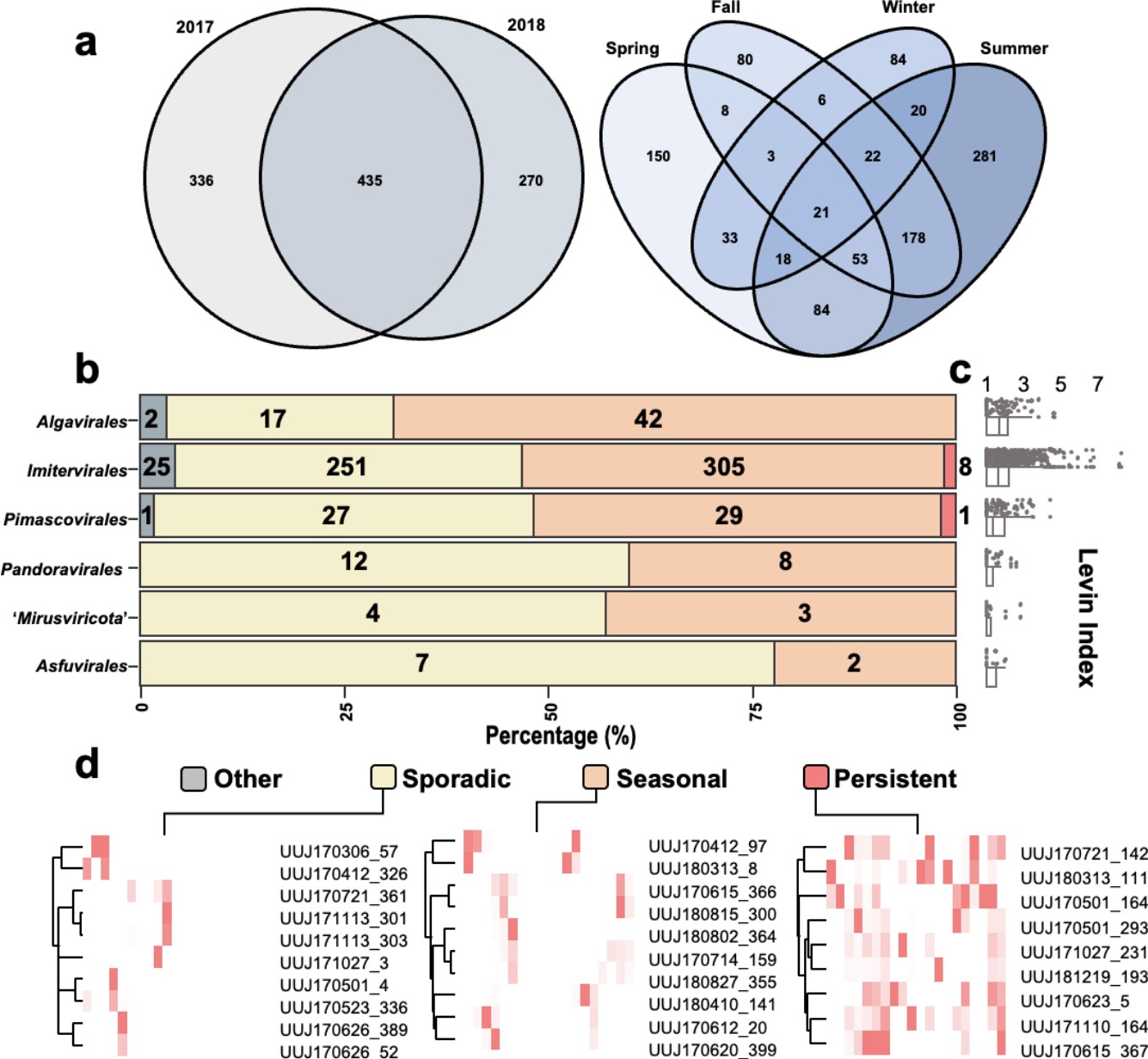

**FIG 4** Three niche categories of giant virus populations. (a) Venn diagrams showing the breakdown of MAG occurrences in different periods. (b) Distribution of MAGs in the niche categories for the six major lineages. (c) Levins' index of each viral lineage. (d) Abundance heatmap for the persistent populations (left), seasonal populations (middle), and sporadic populations (right). The columns represent 20 months from 2017 to 2018, and the rows represent the randomly chosen MAG examples.

categories (Fig. 5d). The pairwise microdiversity similarity of both categories exhibited a yearly cyclical pattern with a peak at approximately 365-day intervals. However, for the viral populations in the other two ecological groups (sporadic or other), a tendency of no or weak recovery was observed. Subsequently, we identified two example patterns of microdiversity dynamics for individual populations in the persistent group (Fig. 6a and b; Fig. S10 and S11). In certain populations, temporal shifts in allele frequencies were observed, with many single nucleotide variant (SNV) sites being dominated by a single allele at specific time points (e.g., UUJ170721_142) (Fig. S10). UUJ170721_142 also demonstrated a clear year-round recovery (Fig. 6b), with seasonal ecotypes (subpopulations) that shared similar microdiversity within the same seasons (Fig. 6c). Despite the year-round recovery, the ND values for this viral population in each month showed no

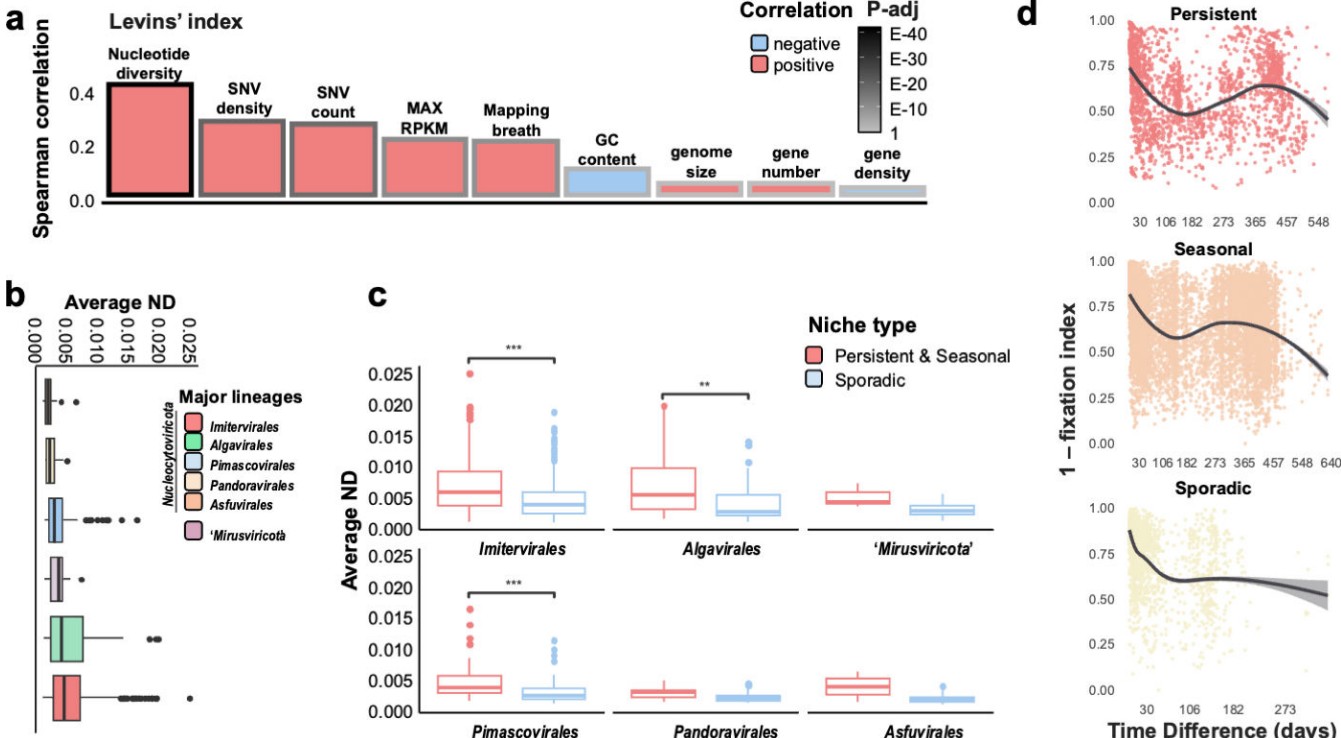

**FIG 5** Persistence and microdiversity. (a) Correlation between niche breadth (Levins' index) and features of giant virus genomes. ND, SNV density, and count of SNV sites were obtained by InStrain v1.0.0. The frame color of the bar plots represents the adjusted *P* value (Benjamini–Hochberg procedure). (b) Average ND of each lineage. Lineages are sorted by median values. (c) Comparison of average ND between generalists (persistent and seasonal) and specialists (sporadic) of each viral lineage. The Wilcoxon rank-sum test was used to determine the significance of comparison. For box plots, center lines show the medians, box limits represent the 25th and 75th percentiles, whiskers extend 1.5 times the interquartile range from the 25th and 75th percentiles, and outliers are represented by dots. (d) Pairwise microdiversity similarity calculated over a 20-month interval that compares the microdiversity of each MAG from every two samples. The pairwise microdiversity similarity level was estimated by (1 − fixation index). The plot is drawn for the viral MAGs of four niche categories: persistent, seasonal, sporadic, and other, respectively. Locally estimated scatterplot smoothing (LOESS) is used to demonstrate the tendency.

relationship with the ecotype clusters (Fig. 6b). Moreover, certain viral populations exhibited relatively stable allele frequencies throughout the months of their occurrence across the 2 years of analysis (e.g., UUJ170623_5, *Pimascovirales*) (Fig. 6a and b; Fig. S11). Additionally, some giant viruses displayed a high degree of variation, and many of the populations from different samples had alleles that differed from the ones in the reference MAG. One example of this pattern was observed in UUJ180313_111, which is phylogenetically close to Organic Lake *Phycodnavirus* (Fig. S10 and S11).

## DISCUSSION

Metagenomics has largely improved our understanding of giant viruses by revealing their distribution across various biomes worldwide using data sets assembled from global samples (28, 30, 31, 33). The usage of genomes from metagenomic assemblies is widely accepted, but different approaches can lead to different interpretations (34). This is particularly crucial for virus studies because viruses exhibit high diversity and there are only a few reference genomes available compared with environmental data (35). A viral MAG is a consensus that masks microdiversity and may not represent any specific genotype in the environment. To deeply analyze viral genomes, such as diversity at the intra-species level, it is critical that the representative genome of each population is high quality. One of the best solutions is manually curating the viral bins using interactive metagenomic tools, like Anvi'o (28, 36). However, this approach demands considerable time, labor, and expertise. To improve research efficiency and reproducibility, there is a need for a pipeline that includes automated curation and refinement processes to

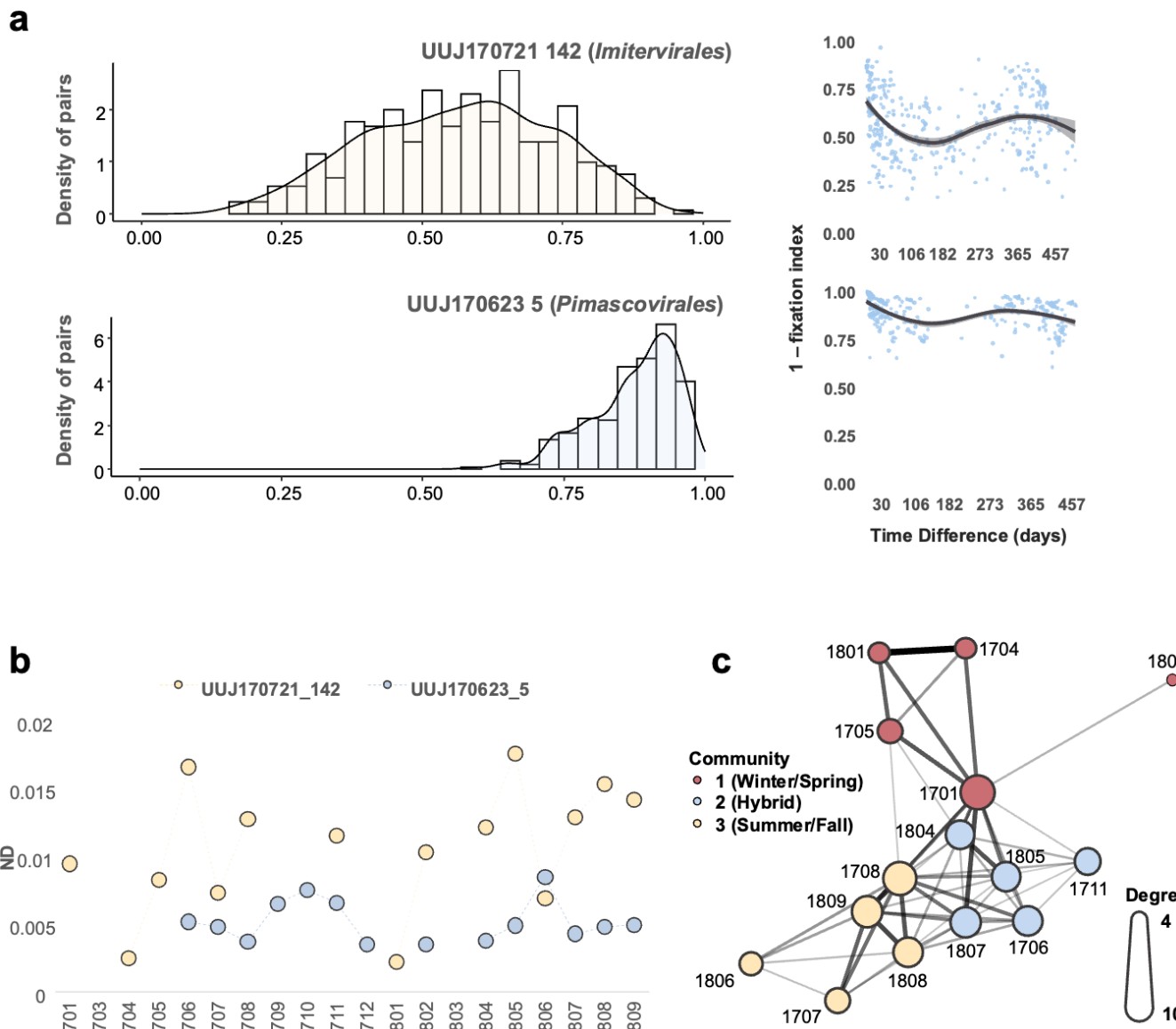

**FIG 6** Temporal dynamics of SNV allele frequencies within different populations. (a) Histogram of similarity values (1 − fixation index) for two MAG samples (left panel). Pairwise microdiversity similarity comparing the microdiversity of each MAG between every two samples of three MAGs (right panel). The pairwise microdiversity similarity level was estimated by 1 − fixation index. LOESS was used to show the trend of year-round recovery. (b) ND dynamics of the two MAGs across 20 months. (c) The network of the pairwise monthly microdiversity similarity of UUJ170721_142 was based on similarity value (1 − fixation index), which is represented by edge width. The size of the nodes represents the degree of the node, and the color represents identified communities. Community search was performed using the R package "igraph" with the function "cluster_louvain." The structure of the network was determined by the "igraph" package with the function "layout_with_fr" and visualized using the "ggraph" package. Only similarities greater than 0.5 were used for the network analysis.

substitute manual checking. To address this, we developed a dedicated metagenomic pipeline for recovering MAGs of giant viruses (Fig. 1), which specifically focuses on removing potential contamination from cellular organisms and eliminating chimeric bins (Fig. 1). Subsequently, to overcome the shortage of reference genomes, we developed a phylogeny-informed quantitative assessment approach based on the principle that evolutionarily related viruses tend to have similar gene contents. Validation using this novel method demonstrated that most of the MAGs generated in this study were high quality, which indicated that the pipeline was efficient for quality control. The MAGs in this study covered all known main lineages (Fig. 2) other than *Chitovirales*, which are not widely distributed and abundant in marine environments (37). Moreover, the automatic

pipeline generated nearly complete mirusvirus genomes (28), which demonstrated that the giant virus screening threshold had high sensitivity for detecting novel giant virus lineages. Overall, our genome data set has high reliability for microdiversity studies. Furthermore, the pairwise ANI pattern of giant viruses (Fig. S3) was similar to that observed in bacteria (27), which supported the concept of "species" for giant viruses, with 95% ANI as an approximate species boundary as suggested by previous studies (38, 39).

Uranouchi Inlet, Japan, is a semi-enclosed eutrophic inlet with high biodiversity, from unicellular organisms to large animals (Fig. S1) (40, 41). A previous study (23) identified year-round recovery of the *Imitervirales* community based on amplicon sequence variants of a single marker gene, DNApolB, in Uranouchi Inlet. Another recent study revealed clear seasonal dynamics of giant virus communities in the photic and aphotic layers of a freshwater lake (42). Although a few additional studies have explored the temporal dynamics of giant viruses (43–45), no study has addressed temporal dynamics of the microdiversity of the whole giant virus community. In this study, we characterized the temporal dynamics of the giant virus community across all environmental giant viral lineages. A consistent year-round recovery was observed throughout 2 years (Fig. 3c and d). In addition to the five orders of the phylum *Nucleocytoviricota*, we also revealed the seasonality of two families (M1 and M2) that belong to a recently discovered phylum, "*Mirusviricota*" (Fig. S6b); this supports a ubiquitous distribution of mirusviruses in marine environments (28). Overall, only a few MAGs ($N = 21$) were present across all seasons (Fig. 4a), which indicates that most of the giant viruses have seasonal preferences. To clearly demonstrate the niche breadth, or ecological strategy, we categorized the viral MAGs into one of three categories: persistent, seasonal, or sporadic (Fig. 4b). The proportion of niche categories varied across lineages, which likely resulted from their distinctive host ranges, as viruses can only thrive when their hosts thrive. For example, a large proportion of seasonal populations was observed in *Algavirales* (Fig. 4b), which may be because they mainly infect algal species exhibiting seasonality (46). On the contrary, *Pimascovirales* also showed abundance peaks across 2 years (Fig. S6), but they displayed more sporadic occurrence than *Algavirales* (Fig. 4b). This may be primarily because pimascoviruses infect large animals, such as fishes (47), and thus are likely carried to Uranouchi Inlet by these swimming organisms. Overall, similar to the heterogeneity observed in their spatial distributions (16, 29), different lineages of giant viruses also exhibited temporal variations in their distributions. The presence of persistent and seasonal giant virus populations primarily contributed to the year-round recovery of the viral community (Fig. 3 and 4). More importantly, we observed that the viral populations of persistent and seasonal categories also exhibited year-round recovery in their intra-population genetic structure. Therefore, seasonal changes were seen at both viral community and population levels. This demonstrates that the dynamics at the population level, such as seasonal ecotypes (Fig. 6c; Fig. S11), may be an important factor that contributes to shaping the seasonal dynamics of the viral community.

Our study revealed a trend that sporadic and persistent giant viruses showed comparatively low and high microdiversity, respectively (Fig. 5; Fig. S8). Theoretically, the level of microdiversity positively correlates with the effective population size (the number of individuals that effectively participate in producing the next generation) under the assumption that mutation rates are comparable across populations and most of the variations are neutral (48, 49). Reduced effective population size amplifies the impact of genetic drift and leads to low diversity, whereas elevated effective population size allows for accumulation of a higher level of neutral mutations within the population and enhances the efficiency of natural selection (50). Notably, similar trends between the level of genetic variation and ecological dynamics have been reported for prokaryotes and eukaryotes in aquatic environments (6, 51–53). Based on the above theoretical framework, those studies hypothesized that increased genetic diversity in cellular organisms was associated with adaptability to specific micro-niches, defense against

viruses (51), stability in abundance (52, 53), and lack of recent population bottleneck (52, 53).

Because viruses are obligate parasites of their hosts, specific factors could account for the relationship between the microdiversity and persistence of viruses, and explain why sporadic viruses rarely show high microdiversity and persistent viruses rarely show low microdiversity (Fig. 7). Viral particles lose infectivity over time. For example, in relatively severe light conditions, the average loss rate of viral infectivity was 0.2 per hour (54). Therefore, sporadic viruses (such as viruses associated with blooming algae) may undergo the bank model of virus–host interactions (55) and experience a genetic bottleneck due to an extended period of inactivity, which leads to microdiversity loss (Fig. 7). Particularly for blooming sporadic viruses, their large populations grow from a small subset of the seed bank and expand rapidly over a short period. As a result, their genetic diversity is supposed to be lower. In contrast, for viruses to void decay and to persist in an environment, they need to recurrently infect hosts that also persist (11, 56). Based on previous observations (6, 51–53), these persistent hosts tend to possess a large effective population size, which results in a higher rate of fixation of advantageous mutations, including defense mechanisms against viruses (57). In this situation, viruses with large population sizes are advantageous because they can rapidly acquire advantageous traits to survive under the severe virus–host arms race. Persistent viruses

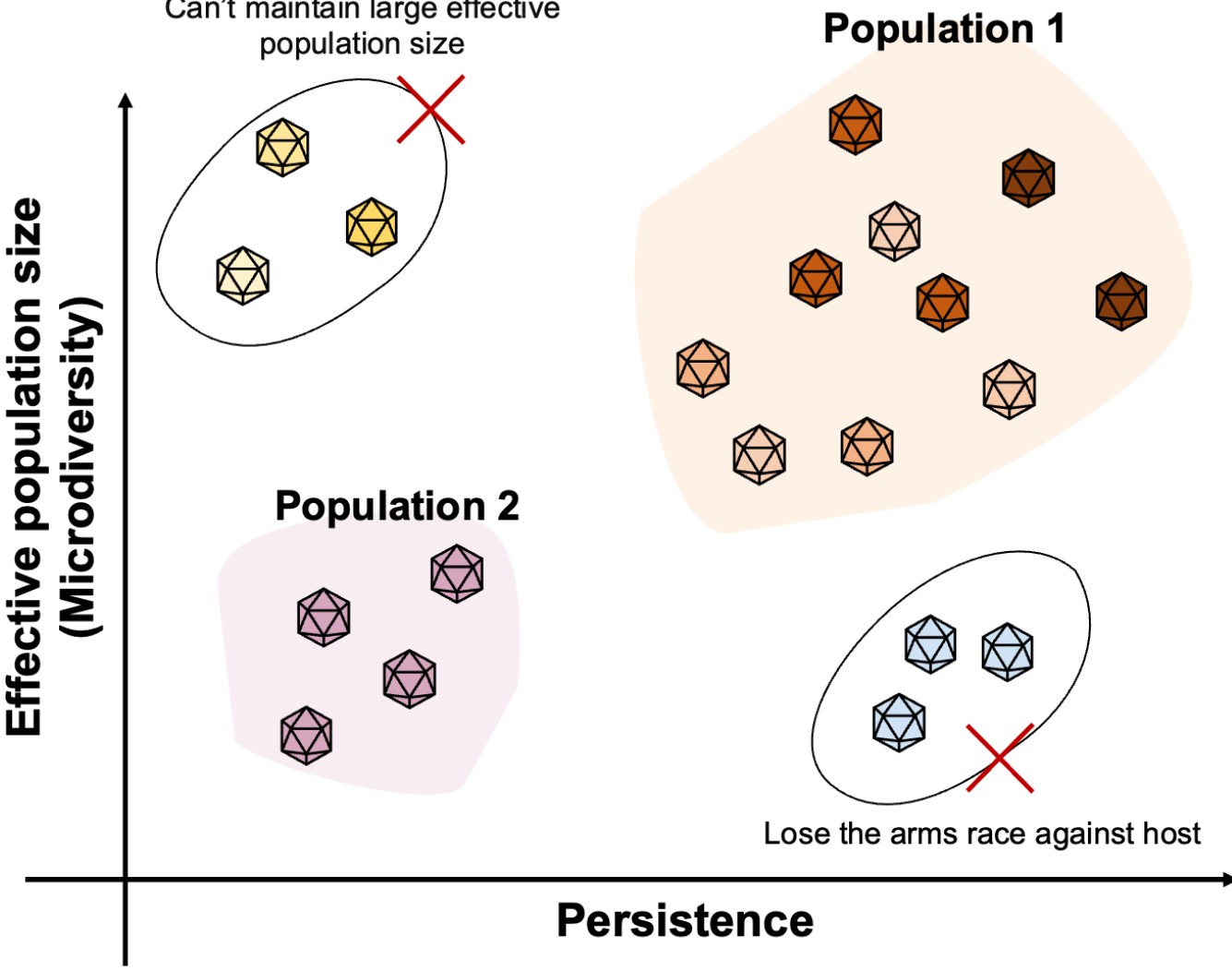

**FIG 7** Summary of the relationships among microdiversity, effective population size, and persistence of giant viruses.

with a small effective population size may become losers in this co-evolutionary arms race.

However, testing of our hypothesis is limited because we only analyzed virus metagenomics during a 20-month observational period. Moreover, in addition to virus–host interactions, competitive exclusion during co-infection might also influence the differences in microdiversity levels within giant virus populations. It is also unclear how or where sporadic viruses can maintain their potential activity without proliferation for an extended period of time. Another limitation is that the detection of viruses using read mapping has relatively low sensitivity due to the dominance of high-abundance cellular organisms and bacterial viruses, which can mask and underestimate the presence of rare lineages. Other types of microdiversity, such as structural variations, are also significant for giant viruses, as they possess extensive gene repertoires that could contribute to their adaptability and evolution. For example, a clear structural variation of the mirusvirus longest open reading frame was observed in our results (Fig. S4b). In future studies, these mechanisms will be further analyzed. Despite these limitations, this research represents the first comprehensive analysis of temporal dynamics of microdiversity in giant viruses. The observed lineage-specific microdiversity provides a novel perspective on the varied ecological and evolutionary processes that affect viral lineages. This study both highlights the critical role of virus–host interactions in shaping the dynamics of giant virus populations and establishes an essential framework for understanding the intricate relationships between these distinct viral entities and their host communities. Populations of giant viruses with high microdiversity may enhance host resource specialization, potentially benefiting nutrient recycling and promoting long-term system stability.

## MATERIALS AND METHODS

### Sample collection, DNA extraction, and sequencing

The water samples used in this study were the same as the ones in a previous study (23) (Table S1). Briefly, seawater samples were collected from three adjacent sites, labeled as "J" (33°25′43.2″N, 133°22′49.5″E), "M" (33°25′60.0″N, 133°24′38.3″E), and "F" (33°26′33.6″N, 133°24′41.8″E), in Uranouchi Inlet, Kochi Prefecture, Japan (Fig. S1a). In total, 42 samples were obtained over 20 months from 5 January 2017 to 25 September 2018 (Fig. S1b). Because the sampling strategy focused on capturing bloom events, a higher sampling frequency was performed during the summer months. Seawater samples (10 L for each sample) were collected from a depth of 5 m and subsequently transported to the laboratory for filtration. The samples were sequentially filtered through 3.0 and 0.8 µm filters (diameter 142 mm, Merck, Darmstadt, Germany), followed by a 0.22 µm filter through a Sterivex filtration unit (Merck, Darmstadt, Germany). After filtration, the filters were stored at −80°C until DNA extraction.

DNA extraction was performed in October 2020 using an in-house protocol (58). Frozen 0.22 µm filters, which included organisms and viruses collected in 0.22–0.8 µm size fractions, were transferred to 1.5 mL microtubes containing 0.1 mm glass beads (0.2 g) and then filled with xanthogenate buffer (1 M Tris-HCl, 0.5 M ethylenediamine-tetraacetic acid [EDTA], 5 M ammonium acetate, 10% potassium xanthogenate, 10% SDS, sterile water). Bead beating (Taitech, Beavercreek, OH, USA) was used to lyse cells and virions, followed by a 60 minute incubation at 70°C to increase DNA yield. Glass beads were removed from the mixture after centrifugation. Then, 600 µL isopropanol was added to the supernatant and mixed. The precipitated DNA was purified with a NucleoSpin gDNA Clean-up Kit (Macherey-Nagel, Düren, Germany) and then dissolved in a Tris-ethylenediaminetetraacetic acid (Tris-EDTA) buffer. A Qubit 4 Fluorometer (Invitrogen, Carlsbad, CA, USA) and the 4150 TapeStation system (Agilent, Santa Clara, CA, USA) were used to measure the quantity and quality of yielded DNA. Extracted DNA was stored at −20°C until sequencing.

Of the 42 extracted DNA samples, 27 were sequenced by Kyushu University (Fukuoka, Japan) and 15 were sequenced by Macrogen Japan (Tokyo, Japan). Both facilities used the same sequencing kits and platforms. The sequencing library was prepared using TruSeq Nano DNA Kit (Illumina, San Diego, CA, USA; insert size, 350 bp), and Illumina NovaSeq 6000 was used to sequence the DNA in paired-end mode (2 × 151 bp, 45 Gbp data for each sample). Quality control of raw sequencing reads was conducted using FastQC v0.11.8 (59), followed by trimming with Trimmomatic v0.39 (60) for adaptors and low-quality reads and fastp v0.20.1 (61), which specifically addressed poly-G tail issues in NovaSeq sequencing. Forty-two metagenomes are available at the DNA Data Bank of Japan (DDBJ) under the BioProject Accession PRJDB12848.

## Generation of giant virus MAGs

The overall workflow for generating giant virus MAGs is outlined in a schematic diagram (Fig. 1; Fig. S2). Briefly, the pipeline comprises the following steps:

### Individual metagenome assembly and binning

For each metagenome, trimmed reads were assembled into contigs using MEGAHIT v1.2.9 (62) in "meta-large" mode. Bowtie2 v2.4.5 (63) mapped reads to assembled contigs longer than 1 Kbp, which were then sorted and converted to BAM files using samtools v1.16 (64). The jgi_summarise_bam_contig_depths script of MetaBAT2 (65) was used to calculate the average mapping depth for each contig. Contigs were clustered into bins using MetaBAT2 v2.12.1 (65), a binning tool that uses both coverage depth and normalized tetra-nucleotide frequency information. Contigs shorter than 2,500 bp were excluded from the binning, and the default MetaBAT2 parameters were employed. For each of the 42 metagenomes, a corresponding set of bins was generated (Fig. 1a).

### Giant virus screening

Previous genome-resolved metagenomic studies for giant viruses have used a fixed set of markers or a single marker gene to detect giant viruses (28, 30, 31). However, these methods are limited by the incompleteness of giant virus MAGs, which may result in overlooking some giant virus genomes or in false-positive detection. A correlation pattern was previously observed between genome size and number of core genes (66). Therefore, we used a core gene density index to screen potential giant viruses. Briefly, we selected 20 marker genes of Nucleo-Cytoplasmic Virus Orthologous Groups (NCVOGs), which are universally or nearly universally present across known giant virus families (67). Then, we assigned weights to 20 NCVOGs according to conservation levels in individual families (from 0 for absence to 1 for the conservation across lineages). Then, the index was calculated using the following equation:

$$Density\ index = \frac{\sum_{k=1}^{20} weight_k}{log10(genome\ size) - 4}$$

To assess the effectiveness of this metric, we compiled a genome database comprising 205 reference giant virus genomes and 6,497 cellular genomes ($N_{archaea}$ = 334, $N_{bacteria}$ = 6,114, $N_{eukaryota}$ = 49) downloaded from the Kyoto Encyclopedia of Genes and Genomes (KEGG) database in June 2019. This density index effectively distinguished giant virus genomes from those of cellular organism genomes (Fig. S2a). Furthermore, our study revealed a distinct gap in the indices of all raw bins created by MetaBAT2, where the gap (corresponding to a density index = 5.75) was the same as the one that discriminates reference viral genomes from cellular genomes (Fig. S2c). Therefore, we used the core gene density index to screen for potential giant viruses.

## Refinement of giant virus MAGs

To improve the quality of the preliminarily screened giant virus MAGs, we employed a rigorous quality control pipeline comprising four principal steps (Fig. 1b):

### Verification of giant virus-specific traits in contigs

To assess giant virus characteristics, we used three taxonomic identification tools: Viralrecall (-c flag), VirSorter2 (--include-groups "dsDNAphage,NCLDV,RNA,ssDNA,lavida-viridae"; "NCLDV" as the target), and CAT v5.2.3 (Diamond NCBI nr protein database; "Nucleocytoviricota" as the target). In addition, a hmm model built with 149 NCVOGs was also used to detect any giant virus signals in the contigs (HMMER3, E−10). Depending on how many out of the three taxonomic identification tools annotate contig as "NCLDV," the contig would first be given the giant virus characteristic score from 0 to 3. Then, as long as there is one gene hit detected against the 149 hmm models, the contig was given 1 characteristic score. In this way, all the contigs from the putative bins were given a giant virus characteristic score from 0 to 4.

### Removal of non-giant virus MAGs from the screened collection

A bin was considered to have no or weak giant virus features if it lacked any of the five NCLDV hallmark genes (MCP, PolB, TFIIB, TopoII, and A32) and had over 90% of its contigs assigned a giant virus characteristic score of "0." These bins were removed.

### Decontamination of viral MAGs to eliminate non-viral sequences

The remaining giant virus bins filtered from the previous step still contained a small proportion of contigs with a giant virus characteristic score of "0," indicating potential contamination due to misbinning. Contigs with a giant virus characteristic score of "0" were removed from the bins.

After removing non-giant virus MAGs, we performed a second-step decontamination process by identifying and removing outlier contigs. Contigs were flagged as outliers based on coverage if they fell outside the range of $Q1 − 1.5 \times IQR$ or $Q3 + 1.5 \times IQR$ (where IQR is the interquartile range for contig coverage within the bin). For tetranucleotide frequency, principal component analysis (PCA) was used, and contigs with a PC1 value outside the range of −2.5 to 2.5 standard deviations were identified as outliers. Contigs classified as outliers based on either criterion and having a characteristic score of "1" were considered non-giant virus contigs and removed.

### Resolution of chimeric bins to separate mixed-lineage bins

Bins with a coverage coefficient (standard deviation/mean) of variance greater than 1 and containing multiple hits for giant virus single copy genes were flagged as potential chimeras. To resolve these, we clustered contigs within these bins based on coverage and tetranucleotide frequency using hierarchical clustering. Those bins were split into two primary clusters, and each cluster was evaluated for giant virus characteristics, including minimum cluster length (>40 kb), presence of single-copy marker genes, high giant virus characteristic scores (>50% of contigs scoring "4"), and consistent species-level taxonomy annotations. In the Uranouchi study, none of the bins met all the criteria for delineation, although the method has been successfully applied in other data sets.

A visual summary of the quality control procedures is depicted in the schematic representation (Fig. 1b).

## Deduplication of MAGs

To construct a representative set of genomes, we used the dereplication tool dRep v3.2.2 (68) to remove redundant MAGs (parameter: --S_algorithm ANImf --ignoreGenomeQuality -pa 0.9 -sa 0.95) (Fig. 1c). dRep performs clustering of MAGs by comparing pairwise

ANI, using Mash for initial rapid screening with a minimum ANI threshold of 90% to group closely related genomes. For more refined and accurate clustering, ANImf was applied with a higher minimum ANI threshold of 95%. The threshold was established based on the distribution of ANI values across all MAGs (Fig. S3), which showed that a 95% ANI value represented a practical boundary for population delineation. In addition, the 95% ANI value has been used as the demarcation criteria to define the viral species for medusaviruses (39) and imiterviruses (38). With the parameter "--ignoreGenomeQuality," dRep assigned scores to MAGs based on N50, genome size, and centrality. The highest-scoring MAG in each cluster was selected as the representative genome of a giant virus population. Finally, 1,065 giant virus MAGs were obtained after deduplication and were used as representative species-level giant virus genomes.

An assessment of the performance of this pipeline is included in the supplementary information.

## Taxonomic classification

Taxonomic classification was determined by referring to a phylogenetic tree that incorporated the 1,065 generated representative species-level MAGs and 205 giant virus reference genomes. This tree was based on a concatenated alignment of the three hallmark genes in the viral informational module (RNApolA, RNApolB, and DNApolB), which comprehensively reveal virus evolution. The hallmark genes were detected using a python program, "ncldv_markersearch" (31). Multiple sequence alignments of the hallmark genes were performed using MAFFT v7.505 (69), followed by concatenation of the three genes. Then, unconserved positions were removed using TrimAl v1.4.1 (70) with a gap threshold of 0.1. A phylogenetic tree was constructed using IQ-TREE v2.2.0 (71) with the LG+F+I+G4 model, which was recommended in a previous study (72). For *Mirusviricota*, an additional HK97-fold major capsid protein tree was reconstructed.

## Phylogeny-informed MAG assessment (PIMA)

In this study, we developed an approach, PIMA, to assess the quality of giant virus MAGs. This approach was designed to overcome the limitations caused by a lack of reference genomes, which impacts the accuracy of quality assessments for giant virus MAGs. This approach requires a guide tree of giant viruses that has been rerooted in accordance with the latest taxonomic classifications and evolutionary scenarios (72, 73). Then, the relative evolutionary divergence (RED) values were calculated to classify taxonomic levels for each clade (e.g., order and family) (72). Within a specific clade, MAG genes were annotated with orthologous groups (OGs); then, core genes in this clade were defined as those identified in more than 50% of the genomes in the clade. We assessed MAG consistency and redundancy using the following equations:

$$Consistency = \frac{number\ of\ core\ genes\ in\ a\ MAG}{number\ of\ core\ genes\ in\ the\ clade}$$

$$Redundancy = \frac{number\ of\ redundant\ genes\ in\ a\ MAG}{number\ of\ core\ genes\ in\ a\ MAG}$$

Redundant genes in a MAG are defined as genes with more copies than the mode copy number (the most common number of copies) for the given genes across all MAGs in the evaluated lineage.

In our study, we adopted a RED value of 0.65 as the threshold for clade definition, which corresponded to the level of viral genus or family (72). MAG quality within each clade was assessed using the above equations within the clade. The consistency and redundancy distribution of the MAGs is shown in Figure S4a. The quality was also assessed by CheckV v1.0.1 by concatenating all contigs and using an "end_to_end" mode (74).

## Population and community dynamics

To assess the time-series dynamics of giant viruses, metagenomic reads were back mapped to 1,065 representative MAGs using Bowtie2 v2.4.5 (63). RPKM was employed to normalize and profile the abundance of giant virus MAGs using CoverM v0.4.0 (75). MAGs were defined as being present in a sample only if they exhibited a coverage breadth (proportion of genomes with at least one read mapped) exceeding 50% of the whole MAG size. Then, we defined "abundant" giant viruses in one sample based on the contribution of the viral MAG to the viral community diversity of the sample, measured by the Simpson index (76). For each sample, the initial step involved calculating the overall Simpson diversity index across all giant virus MAGs. Subsequently, these MAGs were sorted by their relative abundance (RPKM) in descending order. Viral MAGs were considered abundant if, once cumulated, they represented the top 80% of the whole sample diversity based on the Simpson index values.

To assess the time-series dynamics of giant viruses at the community level, the Sørensen–Dice dissimilarity measure was employed to compare community compositions at different times using the vegdist function in the R package "vegan" (77) (method = "bray," binary = T). Because of the uneven frequency of sampling across months, we binned the 42 metagenomes into monthly intervals ($N = 20$) and calculated the averages of all variables for the samples within the same month. This strategy was supported by the observation that samples from the same month exhibited the most similar compositions.

## Genetic diversity of viral populations

The back-mapping files were used as input for InStrain v1.0.0 (78), which facilitated the generation of SNV profiles on both the genome and gene scales. SNVs were exclusively identified at sites with a minimum coverage depth of 5×. The ND, an index of genetic diversity within a population, was calculated according to the method described by Nei and Li in 1979 (79):

$$1 - \left( f_A^2 + f_C^2 + f_G^2 + f_T^2 \right)$$

In the equation, $f$x refers to the frequency of the nucleotide X (A, C, G, or T) at a given nucleotide site. The genome-wide ND value was calculated as the average across all SNV sites within a genome.

In addition to ND, SNV/Mb (the number of SNV sites per million base pairs) was used as an additional metric for assessing microdiversity. The microdiversity analyses focused on abundant viral MAGs because a small number of reads is uninformative and microdiversity metrics for rare MAGs may be unreliable.

## Definition of niche categories

Each giant virus MAG that had appeared before the second winter (December 2017) was classified into one of three niche categories based on its occurrence pattern: persistent, seasonal, or sporadic. The classification criteria for each MAG were based on their occurrence pattern within 8-month sliding windows (1-month step size). The first window spanned January 2017 to August 2017, and the final window covered February 2018 to September 2018. We excluded MAGs that were exclusively present in 2018 and absent in 2017. The definitions of the three niche categories were as follows:

### Persistent

A MAG was classified as "persistent" if it appeared in more than four consecutive months within at least one of the 8-month windows.

## Seasonal

A MAG that was not persistent was classified as "seasonal" if its appearance in 2017 was exclusively in the months that could be entirely covered by at least one 8-month window of 2017, reappeared in the same monthly window of 2018, and did not appear in the remaining months outside that window in 2018. For example, a MAG that appeared in January and September in 2017 was excluded from being classified as seasonal.

## Sporadic and other

A MAG that did not follow the patterns of persistent or seasonal was classified as "sporadic" or "other." Specifically, if the MAG only appeared in 2017, it was labeled as sporadic. If a MAG appeared in both 2017 and 2018, it was classified as other ($N = 28$).

Additionally, the Levins' index, which represents the niche breadth (B) of a giant virus (23), for each giant virus was calculated by the "spaa" package in R, which used the following formula:

$$B_j = \frac{1}{\sum\limits_{i}^{n} p_i^2}$$

In the equation, $p_i$ is the fraction of the relative abundance of virus $j$ in month $i$ out of the sum of the relative abundances of virus $j$ among all months.

## Fixation index

The fixation index is a measure of population differentiation due to genetic structure. It is frequently estimated from single nucleotide polymorphisms. To estimate the fixation index, we followed a previous pipeline (80) and first calculated the distance ($pi$) between two samples for a given MAG at an SNV site as follows:

$$pi(a, b) = a_A\left(\sum_{i=1}^{3} b_{non-A}\right) + a_C\left(\sum_{i=1}^{3} b_{non-C}\right) + a_T\left(\sum_{i=1}^{3} b_{non-T}\right) + a_G\left(\sum_{i=1}^{3} b_{non-G}\right)$$

Here, $a$ and $b$ are vectors representing the nucleotide frequencies of one SNV site of a genome in samples $a$ and $b$, respectively. For example, if there are 30 "A," 0 "C," 10 "T," and 20 "G" mapped at the same site, the vectors for this site will be (1/2, 0, 1/6, 1/3).

Then, the fixation index was calculated using the average distance $pi$ with the following equation. A genome-wide $pi$ value was calculated as the average of the $pi$ values of all SNV sites shared by two genomes:

$$fst(a, b) = 1 - \frac{\left(\frac{\overline{pi(a,a)} + \overline{pi(b,b)}}{2}\right)}{\overline{pi(a, b)}}$$

The genetic structure similarity was measured by $1 - fst$.

## Statistical analysis

The Wilcoxon rank-sum test was employed to identify significant differences in nucleotide diversity, coverage, and SNV/Mb between categorical groups. $P$ values were corrected using the Benjamini–Hochberg procedure in R, and adjusted $P$ values $< 0.05$ were considered significant. Visualization was carried out using R package "ggplot2" (81), DiGAlign v2.0 (82), Cytoscape v3.7.1 (83), and iTol v6 (84).

## ACKNOWLEDGMENTS

This work was supported by JSPS/KAKENHI (Nos. 21H05057, 22H00384, 22H00385, 16H06279 [PAGS]), Scientific Research on Innovative Areas from the Ministry of

Education, Culture, Science, Sports and Technology (MEXT) of Japan (Nos. 16H06429, 16K21723, 16H06437), The Kyoto University Foundation, and the Collaborative Research Program of the Institute for Chemical Research, Kyoto University (Nos. 2021-33, 2019-33, 2018-31, 2017-25). Computational work was completed at the SuperComputer System, Institute for Chemical Research, Kyoto University. We thank Mallory Eckstut, PhD, from Edanz (https://jp.edanz.com/ac) for editing a draft of this manuscript.

Y.F. and L.M. performed most of the bioinformatics analyses in this study. J.X. and Y.O. contributed to the bioinformatics analyses. K.N. performed sampling, and L.M. performed DNA extraction. Y.G. and T.H. contributed to sequencing. H.E. and H.O. designed the study. H.E., Y.O., and H.O. co-supervised Y.F. Y.F. generated the initial draft, and L.M. improved it. All authors contributed to the interpretation of data and writing of the manuscript, and all approved the final draft.

## AUTHOR AFFILIATIONS

[1]Institute for Chemical Research, Kyoto University, Uji, Japan
[2]Department of Bacteriology, Faculty of Medical Sciences, Kyushu University, Fukuoka, Japan
[3]Faculty of Science and Technology, Kochi University, Kochi, Japan

## AUTHOR ORCIDs

Lingjie Meng  http://orcid.org/0000-0001-9937-8860
Hiroyuki Ogata  http://orcid.org/0000-0001-6594-377X

## FUNDING

| Funder | Grant(s) | Author(s) |
| --- | --- | --- |
| MEXT \| Japan Society for the Promotion of Science (JSPS) | 21H05057 | Hiroyuki Ogata |
| Kyoto University \| Institute for Chemical Research, Kyoto University (ICR) | Nos. 2018-31 | Hiroyuki Ogata |
| Kyoto University \| Institute for Chemical Research, Kyoto University (ICR) | Nos. 2017-25 | Hiroyuki Ogata |
| MEXT \| Japan Society for the Promotion of Science (JSPS) | 22H00384 | Hisashi Endo<br>Yusuke Okazaki<br>Hiroyuki Ogata |
| MEXT \| Japan Society for the Promotion of Science (JSPS) | 22H00385 | Hisashi Endo<br>Yusuke Okazaki<br>Hiroyuki Ogata |
| MEXT \| Japan Society for the Promotion of Science (JSPS) | 16H06279 | Hiroyuki Ogata |
| Ministry of Education, Culture, Sports, Science and Technology (MEXT) | 16H06429 | Keizo Nagasaki |
| Ministry of Education, Culture, Sports, Science and Technology (MEXT) | 16K21723 | Keizo Nagasaki |
| Ministry of Education, Culture, Sports, Science and Technology (MEXT) | 16H06437 | Keizo Nagasaki<br>Hiroyuki Ogata |
| Kyoto University \| Institute for Chemical Research, Kyoto University (ICR) | Nos. 2021-33 | Hiroyuki Ogata |
| Kyoto University \| Institute for Chemical Research, Kyoto University (ICR) | Nos. 2019-33 | Hiroyuki Ogata |

## AUTHOR CONTRIBUTIONS

Yue Fang, Data curation, Formal analysis, Investigation, Methodology, Validation, Visualization, Writing – original draft, Writing – review and editing | Lingjie Meng, Data curation, Formal analysis, Investigation, Methodology, Validation, Visualization, Writing – original draft, Writing – review and editing | Jun Xia, Data curation, Methodology, Writing – review and editing | Yasuhiro Gotoh, Data curation, resources, Writing – review and editing | Tetsuya Hayashi, Data curation, resources, Writing – review and editing | Keizo Nagasaki, Data curation, resources, Writing – review and editing | Hisashi Endo, Data curation, Supervision, Writing – review and editing | Yusuke Okazaki, Conceptualization, Methodology, Supervision, Writing – review and editing | Hiroyuki Ogata, Conceptualization, Funding acquisition, Methodology, Project administration, Supervision, Writing – review and editing

## DATA AVAILABILITY

Giant virus MAGs utilized in this study can be accessed from GenomeNet at https://www.genome.jp/ftp/db/community/UranouchiGVMAGs/.

## ADDITIONAL FILES

The following material is available online.

### Supplemental Material

**Supplemental Figures (mSystems01168-24-S0001.docx).** Fig. S1 to S11.
**Supplemental Information (mSystems01168-24-s0002.docx).** An assessment of the performance of binning pipeline used in this study.
**Table S1 (mSystems01168-24-s0003.xlsx).** Information of 42 metagenomes in this study.
**Table S2 (mSystems01168-24-s0004.xlsx).** Basic information of 1,065 MAGs.
**Table S3 (mSystems01168-24-s0005.xlsx).** Quality assessment of 1,065 MAGs.

### Open Peer Review

**PEER REVIEW HISTORY (review-history.pdf).** An accounting of the reviewer comments and feedback.

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
