## [Reviewer comments · mSystems]

Genome-resolved year-round dynamics reveal a broad range of giant virus microdiversity

Yue Fang, Lingjie Meng, Jun Xia, Yasuhiro Gotoh, Tetsuya Hayashi, Keizo Nagasaki, Hisashi Endo, Yusuke Okazaki, and Hiroyuki Ogata

Corresponding Author(s): Hiroyuki Ogata, Kyoto Daigaku

Review Timeline:

Submission Date:	August 28, 2024
Editorial Decision:	September 28, 2024
Revision Received:	November 27, 2024
Accepted:	December 3, 2024

Editor: Xiyang Dong

Reviewer(s): Disclosure of reviewer identity is with reference to reviewer comments included in decision letter(s). The following individuals involved in review of your submission have agreed to reveal their identity: Linxing Chen (Reviewer #1); Frank O Aylward (Reviewer #2)

Transaction Report:

DOI: <https://doi.org/10.1128/msystems.01168-24>

Re: mSystems01168-24 (Genome-resolved year-round dynamics reveal a broad range of giant virus microdiversity)

Dear Prof. Hiroyuki Ogata:

Revision Guidelines

Sincerely,
Xiyang Dong
Editor
mSystems

Reviewer #1 (Comments for the Author):

Please find the attached file.

Reviewer #2 (Comments for the Author):

Fang et al present a valuable insight into recurring patterns of giant virus diversity across a 20-month time-course in Uranouchi inlet, Japan. This is a unique study that addresses an important aspect of viral diversity that is often ignored - namely microdiversity. The most important aspect of this study is the demonstration that patterns of microdiversity are often seasonally recurrent. The benchmarked pipeline is also interesting and will undoubtedly be useful in the future. I have some suggestions for expanding on this in the figures to more clearly show these patterns and to improve the clarity of the writing.

In general, detection of viruses in metagenomes using read mapping has fairly low sensitivity owing to the prevalence of high-abundance lineages that mask rare ones. This is something that should be discussed as a caveat, because some of the lineages may be present year-round but just dip below the threshold of detection for some months. This is pertinent to the discussion around lines 191-195. Also, the term "persistent" should be defined more specifically - does a virus need to maintain the same abundance year-round to be persistent, or would a virus that is consistently present but varies greatly in abundance over seasons still be considered persistent? This is a somewhat technical point but worth considering.

Nucleocytoviruses and mirusviruses are large DNA viruses, and as such an important aspect of their microdiversity will be based on the presence and absence of genes and gene copy number, rather than SNVs. In nucleocytoviruses that infect metazoans it has been shown that gene copy number plays an important role in adaptation to hosts (i.e. the "genomic accordion", Elde et al., 2012, Cell) and it is likely that similar processes occur in viruses of protists. It would be useful for the authors to discuss this caveat in their Discussion and potentially consider looking into gene occurrence in future analyses of microdiversity. This will depend on assembly quality of course, and it will require different methods, but it could be inferred based on read mapping data, for example by looking at fragment recruitment plots to identify genomic islands. In my group we did this with a sporadic virus (Fig 7 <https://doi.org/10.3389/fmicb.2021.657471>) - we did not see any islands, but as the authors show here, sporadic viruses associated with blooms have low microdiversity and may not have these islands.

Line 35 - perhaps "biogeochemical cycles" instead of "material cycles"?

Line 37 - " time-series seawater samples collected from" -> "seawater samples collected across a time-series from"

Line 45 - technically selective pressures do not generate microdiversity (only mutation does), so perhaps re-word to "shape" or "influence" instead of "generate"?

Line 58 - although reference 13 does discuss viruses a bit, this paper is mostly about eukaryotic gene expression. A more relevant work that discussed viral gene expression in more detail would be <https://doi.org/10.1128/msystems.00293-21>

Line 63-64 - it is not quite true that most imiterviruses have synchronous cycles - some appear to be quite prevalent year-round across wide swaths of the ocean - see <https://doi.org/10.3389/fmicb.2022.1021923> and <https://doi.org/10.1038/s43705-023-00252-6>. In general these references provide some useful background information on giant virus diversity in the ocean that is relevant to the introduction and the discussion around lines 307-317. For example, the prevalence of imiterviruses, pimascoviruses, and algaviruses was found in marine bioGeotraces data, and seasonality was detected in several abundant NCLDV in Station ALOHA.

Line 99 - a brief explanation of PIMA, including spelling out the acronym, would be appropriate here. Was this method developed in this study? If so it would be worth mentioning here.

It is never really described what ND stands for. I thought it was some measure of niche differentiation at first. In general, these statistics and abbreviations need to be explained as they are introduced in the main text so that the reader does not need to search the methods. Otherwise it is challenging to follow the rationale and major findings of the manuscript.

Figure 3b - it would be useful to have absolute numbers next to the bars, so that one can judge the patterns better. For example, I believe there are many more Imitervirales than Asfuvirales.

Line 211- some description of the Levins index and what it is measuring needs to be provided here.

Figure 4a - was Levin's index correlated to relative abundance? As the authors mention in the discussion, nucleotide diversity would be expected to be positively correlated with population size, so this would be worth checking. I understand that some corrections were made in 239-246 to ensure that the ND estimates were not biased by the depth of sequencing, but it would also be worth correlating with relative abundance. Relative abundance is also likely correlated to niche breadth, because abundant taxa may be above the threshold of detection across multiple seasons. This may explain the observation that "giant viruses with higher persistence levels displayed higher ND".

Figure 4a - what is SNV here? The total number of SNVs?

In Figure 4d it looks like there is a seasonal pattern in both the seasonal and persistent viruses.

The schematic in Figure 6 seems incomplete, or perhaps I am missing the major message. I understand that large populations

are less likely to go extinct, but why do the populations with low persistence and low abundance continue to exist? Presumably the sporadic viruses include *Emiliana huxleyi* virus and other viruses that lead to algal bloom demise - they appear during blooms but then cannot be found at other times. However, they are very abundant for a brief window of time. It is unclear to me how these viruses persist - presumably there must be some reservoir in between blooms. I'm not sure this schematic captures this complexity, but perhaps I am missing a detail.

For Figure 5 I am not sure why these specific viruses were chosen to highlight. Are these patterns broadly consistent across other viruses? Panel C is quite interesting and does a good job of showing seasonally recurring patterns in the microdiversity of this population. If it would be possible to make a figure that had an array of these networks, perhaps divided by ecological category, it might do a nice job of showing the general trends. In my view, this finding is probably the most important aspect of this study.

Frank Aylward

Review comments

for

Genome-resolved year-round dynamics reveal a broad range of giant virus microdiversity

In this study, Fang et al. developed a new analysis pipeline to generate MAGs of eukaryotic viruses from metagenomic datasets. Via comparing against two previously published pipelines, the authors claimed their developed pipeline was the best among these three. With this pipeline, the authors analyzed eutrophic coastal seawater samples collected over 20 months and generated a total of 1065 high-quality genomes, which covered 6 major giant virus lineages/groups. They found distinct dynamics for viral populations that (1) persisted in the community, (2) presented upon season, and (3) were sporadic. They also found that giant viruses with broader niche breadth tended to exhibit higher levels of microdiversity.

I found this manuscript well-organized, and I like that the authors were able to put most of the details in the Supplementary Information, thus readers could probably repeat the analyses. However, some necessary details are likely missing. I have some comments I hope could help the authors for the improvement of the manuscript.

Major comments

1. The authors highlighted the “newly developed computational pipeline” in the abstract. However, this was not described in the results section. Given that (1) viral binning could lead to unexpected contamination into the MAGs, and (2) viral MAGs are the base of this study, the underlying mechanisms and the advantages of the pipeline (over published tools) should be described in the main text. For example, I suggest moving Figure S2 to the main text along with corresponding descriptions. Probably add a paragraph (as the first paragraph of the Results section) to include the main idea and results of the developed pipeline.
2. The authors also developed an approach to evaluate the quality of viral MAGs, which is good. However, it was likely they did not show if they had validated the reliability of the approach by benchmarking.
3. There are several things of potential interest that the authors mentioned in the manuscript without showing more details (see related comments below), which is a pity, as a reader, I would like to know more about them.

Specific and/or minor comments

Line 20, in these viruses -> of these viruses

Lines 45-47, please provide reference(s).

Lines 49-50, could the authors briefly state why?

Lines 76-77, better state the total size of the reads (for example, Gbp), as the length of reads may vary.

Lines 91-93, “we screened 3,082 potential giant virus bins”, “subsequently refined them to enhance their quality”, please describe how?

Line 98, “species-level references”, at what identity percentage, please?

Line 102, as mentioned elsewhere, please describe how to use CheckV on MAGs.

Line 112, how about the remaining 13?

Lines 119-122, any other details about these clades?

Line 148, M1 and M2 were not indicated in Figure 1d. Please add.

Lines 158-160, not sure if this is an overstatement because (1) this is just a sampling of two years, and (2) the dominant lineages between the two summers were different. (after reading Figure 2 and related descriptions, I think this is probably ok)

Lines 168-171, where is the corresponding table or figure?

Line 194, please describe how to define the presence of a virus in a given season, should it be detected in one month or all three months?

Line 410, the project of PRJDB12848, is not available yet. Please confirm.

Figures

Figure 3, there are two "C"s in the figure, should one of them be a "D"? I suggest using different colors in subfigure a from other subfigures. For text legend, I think "line" and "column" probably are better, instead of "x-axis" and "y-axis".

Figure S2, (d) varification -> verification, and the text legend descriptions for subfigures (d) and (e) are missing.

Supplementary_Information

Page 3, "A total of 498 MAGs were generated by the three pipelines. The pipeline used in this study produced 153 MAGs, while the other two pipelines generated 312 and 33 MAGs, respectively (Fig. SI1)." Does it mean the three pipelines obtained no overlapped MAG?

Page 3, As far as I know, CheckV could only evaluate the quality of single sequences, it is unclear how the authors performed the analysis for MAGs, please describe.

Page 4, Figure SI3, it is better to list in the order of "Pipeline1, Pipeline2, This study" as did in Figure SI2.

Page 4, this reviewer may disagree with "making it an ideal choice in giant virus study", for me, I prefer fewer MAGs (pipeline2) with less contaminations.

Page 5, currently it is unclear how the genome consistency analysis was performed and how readers should read the results shown in the figure. Please state.

Blue: Comments

Green: Response

Reviewer #1 (Remarks to the Author):

In this study, Fang et al. developed a new analysis pipeline to generate MAGs of eukaryotic viruses from metagenomic datasets. Via comparing against two previously published pipelines, the authors claimed their developed pipeline was the best among these three. With this pipeline, the authors analyzed eutrophic coastal seawater samples collected over 20 months and generated a total of 1065 high-quality genomes, which covered 6 major giant virus lineages/groups. They found distinct dynamics for viral populations that (1) persisted in the community, (2) presented upon season, and (3) were sporadic. They also found that giant viruses with broader niche breadth tended to exhibit higher levels of microdiversity.

I found this manuscript well-organized, and I like that the authors were able to put most of the details in the Supplementary Information, thus readers could probably repeat the analyses. However, some necessary details are likely missing. I have some comments I hope could help the authors for the improvement of the manuscript.

We sincerely thank the reviewer for their time and insightful comments on our manuscript. We appreciate the positive feedback on the research and organization of the manuscript, as well as the constructive suggestions for improving clarity and content. In response, we have carefully revised the manuscript, incorporating the suggested changes, especially by adding more details about PIMA from the supplementary information to the main text. Additionally, we have included validation results of PIMA in the supplementary materials. We believe that these revisions have strengthened the manuscript and made it more clear and understandable.

Our detailed responses, to each comment, are listed below.

Major comments

1. The authors highlighted the “newly developed computational pipeline” in the abstract. However, this was not described in the results section. Given that (1) viral binning could lead to unexpected contamination into the MAGs, and (2) viral MAGs are the base of this study, the underlying mechanisms and the advantages of the pipeline (over published tools) should be described in the main text. For example, I suggest moving Figure S2 to the main text along with corresponding descriptions. Probably add a paragraph (as the first paragraph of the Results section) to include the main idea and results of the developed pipeline.

Thank you for highlighting this oversight. We agree that a verified pipeline is fundamental for this genome-resolved study. As suggested, we have moved a brief description and plots (Original Fig. S2a-e, Fig. SI4, and Fig. SI5) of the pipeline from the Supplementary Figures and Supplementary Information to the main text, creating a new section at the beginning of Results, named “Pipeline for Generating Giant Virus Genomes” (LN90-LN113).

To maintain the flow, structure, and clarity of the manuscript, we have condensed the information to emphasize only key aspects, such as the pipeline steps and few results

about the comparison with other approaches. Full details are still available in the Supplementary Information, allowing readers to access them as needed. Moreover, we have added more details about the refinement step of giant virus MAGs in the Methods section. Together with all modifications, we hope the revised manuscript could provide more comprehensive and clearer understanding of our pipeline.

2. The authors also developed an approach to evaluate the quality of viral MAGs, which is good. However, it was likely they did not show if they had validated the reliability of the approach by benchmarking.

We apologize for the lack of clarity. Due to the word limit and scope of the main text, we have provided more information and validation about PIMA (the approach used for quality evaluation) in the Supplementary Information, including the concept (Figure SI3; LN 105-116 in Supplementary Information). To validate its reliability, we newly performed a comparison between PIMA and CheckV using the *Tara* Arctic MAG data (Figure SI4), which aligns with the narrative of other parts in the Supplementary Information (LN 116-124 in Supplementary Information). Even though the overall trends showed a correlation between the CheckV completeness and PIMA consistency, the two methods provided different values due to the differences in their algorithms.

To further support the reliability of PIMA, we included its application on reference genomes in the main text as validation. Reference genomes yielded significantly higher PIMA quality values compared to MAGs, with a consistency of 93.33% and redundancy of 6.67%. Although these values are not perfect, as references are expected to be 100% complete and contamination-free, the discrepancy arises from PIMA's algorithm, which evaluates the relatedness of the query genome. For instance, molliviruses, which exhibit the lowest consistency among all references, are closely related to pandoraviruses, leading to lower estimated quality values.

Furthermore, as CheckV is a more widely used and accepted tool, we added quality information of CheckV values in the main (LN 135-137) text to facilitate readers' understanding.

3. There are several things of potential interest that the authors mentioned in the manuscript without showing more details (see related comments below), which is a pity, as a reader, I would like to know more about them.

Thank you for your valuable suggestion. We have carefully reviewed each point, restructured the relevant paragraphs, and added the necessary details to enhance the readability and completeness of the manuscript.

Specific and/or minor comments

Line 20, in these viruses -> of these viruses

Thank you, this phrase has been modified as above (LN20).

Lines 45-47, please provide reference(s).

We have added the citations below (LN47).

<https://pubmed.ncbi.nlm.nih.gov/11077156/>

<https://pubmed.ncbi.nlm.nih.gov/32169939/>

<https://doi.org/10.4319/lo.1999.44.3.0628>

<https://pubmed.ncbi.nlm.nih.gov/38950433/>

Lines 49-50, could the authors briefly state why?

The lack of focus on viral microdiversity is mainly due to limited availability of high-resolution metagenomic data, the complexity of viral communities, and challenges with reliable binning.

This could be explained by a previous paper

(<https://pubmed.ncbi.nlm.nih.gov/31031001/>):

“In nature, viral microdiversity measurements have been limited to marker genes (e.g., genes encoding major capsid proteins), which capture neither community-wide variability nor genome-wide evidence of selection. Recently, deeper metagenomic sequencing and population genetic theory-grounded species delimitations have begun to reveal such microdiversity in microbes, and this has elucidated unknown features of speciation, adaptation, pathogenicity, and transmission.”

We have refined the sentence above in a more positive tune and inserted in the main text.

“Recent metagenomics studies started to investigate the microdiversity of environmental viruses (mainly small bacterial viruses), revealing temporal changes of microdiversity and relationships between microdiversity and biogeography.” (LN49-51)

Lines 76-77, better state the total size of the reads (for example, Gbp), as the length of reads may vary.

The total size has been modified to “1.8 Tbp” (LN 78-79).

Lines 91-93, “we screened 3,082 potential giant virus bins”, “subsequently refined them to enhance their quality”, please describe how?

We have added a brief description in the new section of the Results (LN 96-99) and a detailed explanation of the refinement process in the Methods section (LN432-468).

Line 98, “species-level references”, at what identity percentage, please?

We have added “95% ANI” in the sentence (LN 126).

Line 102, as mentioned elsewhere, please describe how to use CheckV on MAGs.

We concatenated the contigs of the given MAGs as input for CheckV, using the “end_to_end” mode. This information has been added to the Methods section (LN 519).

Line 112, how about the remaining 13?

The remaining 13 genomes are Mirusviruses that we described in the next paragraph. In the revised manuscript, we have mentioned 13 Mirusviruses in LN 145 to avoid confusion.

Lines 119-122, any other details about these clades?

These clades are particularly interesting and worth deeper exploration. For the sake of a concise narrative in this paper, we only added the gene content of these clades, providing the proportion of unique and shared genes within these genomes compared

to giant viruses of other clades. We also provided a core/pan-genome plot for CladeC, which has largest genomes and particularly intriguing (Fig. S5 b, c).

Line 148, M1 and M2 were not indicated in Figure 1d. Please add.

We have added labels for families M01 and M02 to the internal nodes (Fig. 2d).

Lines 158-160, not sure if this is an overstatement because (1) this is just a sampling of two years, and (2) the dominant lineages between the two summers were different. (after reading Figure 2 and related descriptions, I think this is probably ok)

Thank you for highlighting this confusion. We have removed the statement, “which indicated the presence of a consistent annual pattern within the giant virus community,” to reduce the ambiguity.

Lines 168-171, where is the corresponding table or figure?

We have cited Fig. S7a and Fig. 3b for this result (LN 187).

Line 194, please describe how to define the presence of a virus in a given season, should it be detected in one month or all three months?

We define the presence of a virus in a given season when the virus is abundant in at least one month of that season. We have added this definition to the text (LN 200).

Line 410, the project of PRJDB12848, is not available yet. Please confirm.

We have changed the status to “release” on Oct. 17th, 2024.

Figures

Figure 3, there are two “C”s in the figure, should one of them be a “D”? I suggest using different colors in subfigure a from other subfigures. For text legend, I think “line” and “column” probably are better, instead of “x-axis” and “y-axis”.

Thank you for pointing out the mistake. We have corrected the “c” to “d” accordingly. Additionally, we modified the colors of subfigures (a) and (c) for better clarity. Since the colors of (b) and (d) are meant to correspond to each other, we chose to keep them same as before.

We have also updated the legend to according to your suggestion.

Figure S2, (d) varification -> verification, and the text legend descriptions for subfigures (d) and (e) are missing.

We have moved Fig. S2a, S2d, and S2e to the main text and corrected the term “varification” to “verification” (Fig. 1a). Additionally, we have added descriptions for subfigures (d) and (e).

Supplementary Information

Page 3, “A total of 498 MAGs were generated by the three pipelines. The pipeline used in this study produced 153 MAGs, while the other two pipelines generated 312 and 33 MAGs, respectively (Fig. SI1).” Does it mean the three pipelines obtained no overlapped MAG?

A significant proportion of overlap was observed among them based on pairwise ANI. For example, sixteen groups of quadruple genome consensuses were detected (i.e., each group included MAGs from all three pipelines and one GOEV MAG). Given that the total number of bins recovered from Pipeline_3 was 33, the proportion of bin overlap was

sufficient (About half bins generated by Pipeline_3 are represented in all other three datasets).

To clarify, we have removed the sentence “A total of 498 MAGs were generated by the three pipelines” from the supplementary information.

Page 3, As far as I know, CheckV could only evaluate the quality of single sequences, it is unclear how the authors performed the analysis for MAGs, please describe.

Same to a previous comment that we concatenated the contigs of given MAGs as the input to checkV. Use the “end_to_end” mode. This information has been added to the Methods section (LN 519 in the main text).

Page 4, Figure SI3, it is better to list in the order of “Pipeline1, Pipeline2, This study” as did in Figure SI2.

Page 4, this reviewer may disagree with “making it an ideal choice in giant virus study”, for me, I prefer fewer MAGs (pipeline2) with less contaminations.

We have modified the order of the x-axis (Fig. SI5). Our aim is not to claim superior performance of our pipeline but to demonstrate its suitability for supporting microdiversity studies. To avoid misunderstanding, we have removed the sentence “making it an ideal choice in giant virus study” (LN 100 in SI).

Page 5, currently it is unclear how the genome consistency analysis was performed and how readers should read the results shown in the figure. Please state.

We have provided a more detailed explanation of how PIMA works, including the consistency calculation, in the supplementary information (LN 105-124 in SI). The current structure of the supplementary information could make it easier for readers to understand the value of this approach.

Reviewer #2 (Remarks to the Author):

Fang et al present a valuable insight into recurring patterns of giant virus diversity across a 20-month time-course in Uranouchi inlet, Japan. This is a unique study that addresses an important aspect of viral diversity that is often ignored - namely microdiversity. The most important aspect of this study is the demonstration that patterns of microdiversity are often seasonally recurrent. The benchmarked pipeline is also interesting and will undoubtedly be useful in the future. I have some suggestions for expanding on this in the figures to more clearly show these patterns and to improve the clarity of the writing.

We appreciate the constructive feedback from Prof. Aylward. Below is our point-by-point response to each comment, and we are grateful for the opportunity to enhance our manuscript based on these valuable suggestions. In particular, we have expanded the discussion to address potential limitations and outline possible improvements to the study.

In general, detection of viruses in metagenomes using read mapping has **fairly low sensitivity** owing to the prevalence of high-abundance lineages that mask rare ones. This is something that **should be discussed as a caveat**, because some of the lineages may be present year-round but just dip below the threshold of detection for some

months. This is pertinent to the discussion around lines 191-195. Also, the term **"persistent" should be defined more specifically** - does a virus need to maintain the same abundance year-round to be persistent, or would a virus that is consistently present but varies greatly in abundance over seasons still be considered persistent? This is a somewhat technical point but worth considering.

Thank you for highlighting the sensitivity issue related to read mapping. It is indeed challenging to accurately capture the true microdiversity due to the limitations of sequencing depth. We have added a limitation caveat in the discussion section (LN347-350).

Regarding the issue of 'persistent' viruses, it is challenging to distinguish low relative-abundance from undetected ones due to the limitations of sequencing depth and interference from the existence of cellular organisms. So, we only considered the abundant ones for the definition of 'persistent'. In our study, we defined 'presence' of viruses in a sample (and for the niche breadth calculation) based on their contribution to the viral community alpha diversity, measured by the top 80% of the sample's diversity using Simpson's diversity index (LN 526-532). This approach, as we propose, digitizes the frequency (0 or 1) based on their contribution to track the significant dynamics that shape the viral community, while ignoring fluctuations within presence or absence (by treating undetectable or low abundance cases as absence). We have modified the sentence to "the frequency of giant virus MAGs being abundant over a period of 20 months exhibited a right-skewed distribution pattern" (LN 196-198).

Nucleocytoviruses and mirusviruses are large DNA viruses, and as such an important aspect of their microdiversity will be based on the presence and absence of genes and gene copy number, rather than SNVs. In nucleocytoviruses that infect metazoans it has been shown that gene copy number plays an important role in adaptation to hosts (i.e. the "genomic accordion", Elde et al., 2012, Cell) and it is likely that similar processes occur in viruses of protists. It would be useful for the authors to discuss this caveat in their Discussion and potentially consider looking into gene occurrence in future analyses of microdiversity. This will depend on assembly quality of course, and it will require different methods, but it could be inferred based on read mapping data, for example by looking at fragment recruitment plots to identify genomic islands. In my group we did this with a sporadic virus (Fig 7 <https://doi.org/10.3389/fmicb.2021.657471>) - we did not see any islands, but as the authors show here, sporadic viruses associated with blooms have low microdiversity and may not have these islands.

We agree that gene presence/absence and gene copy number variations are crucial aspects of viral adaptation, particularly for large DNA viruses. In fact, our results have demonstrated the significance of this aspect. As shown in the Mirusvirus pair in Fig. S4, a clear structural variation was observed at around 300 Kbp site of the Mirus_Closed_Genome. This variation occurred in the middle of contigs and in a pair of the longest ORFs in same ortholog group (94% amino acid identity and 72.4% coverage of the longer ORF), resulting in the genome detected in the Mediterranean Sea being larger than the one from Japan. We have included this limitation and future direction in the discussion (LN350-353).

For future studies, we plan to investigate the gene repertoire among bins derived from the same MAG species, utilizing long-read sequencing and a mapping approach to detect metagenomic islands. As demonstrated in our previous paper, long-read sequencing has shown highly effectiveness in detecting structural variations [<https://journals.asm.org/doi/10.1128/msystems.00433-22>]. Thank you for the constructive suggestion.

Line 35 - perhaps "biogeochemical cycles" instead of "material cycles"?
We have changed the word based on suggestion (LN 35).

Line 37 - " time-series seawater samples collected from" -> "seawater samples collected across a time-series from"
We have revised the sentence (LN21-22).

Line 45 - technically selective pressures do not generate microdiversity (only mutation does), so perhaps re-word to "shape" or "influence" instead of "generate"?
We choose to use "influence" to replace "generate" (LN 45).

Line 58 - although reference 13 does discuss viruses a bit, this paper is mostly about eukaryotic gene expression. A more relevant work that discussed viral gene expression in more detail would be <https://doi.org/10.1128/msystems.00293-21>
Thank you for the recommendation. We decided to cite both references as they provide valuable insights from different perspectives (LN 59). One offers a comprehensive view of giant virus transcription on a global marine scale, while the other one provides detailed information for giant virus gene expression from a time-series analysis.

Line 63-64 - it is not quite true that most imiterviruses have synchronous cycles - some appear to be quite prevalent year-round across wide swaths of the ocean - see <https://doi.org/10.3389/fmicb.2022.1021923> and <https://doi.org/10.1038/s43705-023-00252-6>. In general these references provide some useful background information on giant virus diversity in the ocean that is relevant to the introduction and the discussion around lines 307-317. For example, the prevalence of imiterviruses, pimascoviruses, and algaviruses was found in marine bioGeotraces data, and seasonality was detected in several abundant NCLDV in Station ALOHA.
Thank you for your helpful suggestion. The previous description, "Generally, the Imitervirales community exhibits synchronous seasonal cycles with eukaryotes and year-round recurrence," referred to the dynamics at the community level. However, as you mentioned, the patterns of individual viruses are more divergent, with many prevalent viruses observed. We have added two citations along with the relevant description to support this.
Specifically, we have added a sentence in the introduction: "Similarly, distinct patterns in the seasonality of individual giant viruses at Station ALOHA were observed." (LN 65-66) to emphasize the variability in individual viral patterns. Additionally, we removed the use of "Generally" to enhance the rigor of the statement (LN 63).

Line 99 - a brief explanation of PIMA, including spelling out the acronym, would be appropriate here. Was this method developed in this study? If so it would be worth mentioning here.

Thank you for the suggestion. We have added a brief description and acronym (LN 127). Additionally, as recommended by Reviewer 1, we have expanded the information about PIMA in the Supplementary Information (LN 105-124, SI), including a detailed explanation of its concept and validation using the *Tara* Arctic data.

It is never really described what ND stands for. I thought it was some measure of niche differentiation at first. In general, these statistics and abbreviations need to be explained as they are introduced in the main text so that the reader does not need to search the methods. Otherwise it is challenging to follow the rationale and major findings of the manuscript.

Sorry for the miss of description, we have added the “nucleotide diversity” in LN 215. Additionally, we have checked that all abbreviations, including ND (Nucleotide Diversity), are properly explained in the main text when they are first introduced.

Figure 3b - it would be useful to have absolute numbers next to the bars, so that one can judge the patterns better. For example, I believe there are many more Imitervirales than Asfuvirales.

We agree that the count of genomes will help in understanding the pattern. Accordingly, we have added the absolute counts of genomes to Fig. 4b to provide a clearer representation of the data.

Line 211- some description of the Levins index and what it is measuring needs to be provided here.

We have revised the sentence to “Then, we investigated factors that are associated with niche breadth (persistence) measured by the Levins’ index (Fig. 5a), which takes into account the number of months a taxon occupies and its relative contributions to respective months, with higher values indicating generalist taxa that are equally abundant across multiple months” (LN 211-214)

Figure 4a - was Levin's index correlated to relative abundance? As the authors mention in the discussion, nucleotide diversity would be expected to be positively correlated with population size, so this would be worth checking. I understand that some corrections were made in 239-246 to ensure that the ND estimates were not biased by the depth of sequencing, but it would also be worth correlating with relative abundance. Relative abundance is also likely correlated to niche breadth, because abundant taxa may be above the threshold of detection across multiple seasons. This may explain the observation that “giant viruses with higher persistence levels displayed higher ND”. Thank you for your insightful suggestion. We agree that examining the relationship between Levins’ index and relative abundance could provide valuable insights. To address this, we updated Fig. 5a and added the correlation coefficient between Levins’ index and maximum relative abundance. From the results, we observed the strength of this correlation is much lower compared to the correlation between Levins’ index and nucleotide diversity (0.20 vs. 0.41). Another support is that the correlation between ND and relative abundance is not strong (Fig. S9b; $R^2 = 0.02$).

It is important to note that relative abundance may not strongly reflect absolute abundance. In theory, ND is largely neutral and strongly correlated with population size, which could potentially related with absolute abundance, but not necessarily in relative

abundance. Therefore, we do not expect a strong correlation between ND and relative abundance.

Figure 4a - what is SNV here? The total number of SNVs?

We used the total number of detected SNV sites, and SNV density is defined as the total number of SNVs normalized by the genome size. We have revised the label to "SNV count" in Fig. 5a to clarify this definition.

In Figure 4d it looks like there is a seasonal pattern in both the seasonal and persistent viruses.

Thank you for pointing this out. Yes, in Figure 5d, we do observe a seasonal pattern in both the seasonal and persistent viruses. We explicitly acknowledged the existence of seasonal patterns in both the seasonal and persistent categories, while the other two ecological groups in LN 243 represent 'Sporadic' and 'Other.' The previous writing might have been unclear.

To avoid confusion, we have modified the statement from "in the other two ecological groups, a tendency of no or weak recovery was observed." to "in the other two ecological groups (Sporadic or Other), a tendency of no or weak recovery was observed." (LN 242-243)

The schematic in Figure 6 seems incomplete, or perhaps I am missing the major message. I understand that large populations are less likely to go extinct, but why do the populations with low persistence and low abundance continue to exist? Presumably the sporadic viruses include *Emiliana huxleyi* virus and other viruses that lead to algal bloom demise - they appear during blooms but then cannot be found at other times. However, they are very abundant for a brief window of time. It is unclear to me how these viruses persist - presumably there must be some reservoir in between blooms. I'm not sure this schematic captures this complexity, but perhaps I am missing a detail.

Thank you for your thoughtful comment. How sporadic viruses (like blooming viruses) can persist is indeed a challenging aspect to explain. Our model suggests that they produce in large numbers, but because the next bloom occurs a year later, there is a strong bottleneck effect due to decay. Also, for blooming sporadic viruses, no matter how large the seed bank is, their large blooming population occurs from a small part of the bank and grows fastly in a short time, so that the genetic diversity is supposed to be lower compared with persistent viruses. This is our current model; however, if we start to speculate about latent infections or the existence of completely unknown reservoirs, it becomes highly speculative and doesn't align well with the simple theory of correlation between ND and niche breadth.

To improve clarity, we have rephrased the discussion as follows: "Therefore, sporadic viruses may undergo the bank model of virus–host interactions and experience a genetic bottleneck that leads to microdiversity loss (Fig. 7)." This has been revised to: "Therefore, sporadic viruses (such as those associated with blooming algae) may undergo the bank model of virus–host interactions and experience a genetic bottleneck due to an extended period of inactivity, which leads to microdiversity loss (Fig. 7). Particularly for blooming sporadic viruses, their large populations grow from a small subset of the seed bank and expand rapidly over a short period. As a result, their genetic diversity is supposed to be lower." (LN 331-336)

We have also added a caveat in the final paragraph of the discussion. We added "It is also unclear how or where sporadic viruses can maintain their potential activity without proliferation for an extended period of time." in LN346-347.

For Figure 5 I am not sure why these specific viruses were chosen to highlight. Are these patterns broadly consistent across other viruses? Panel C is quite interesting and does a good job of showing seasonally recurring patterns in the microdiversity of this population. If it would be possible to make a figure that had an array of these networks, perhaps divided by ecological category, it might do a nice job of showing the general trends. In my view, this finding is probably the most important aspect of this study.

The selection of this MAG for Figure 6 was initially based on an observation of microdiversity dynamics patterns shown in Figure S10. These dynamics vary among nine persistent MAGs (MAGs from other niche categories did not provide sufficient information to capture the dynamics of microdiversity). Some, like UUJ170623_5, show homogeneity within the population, while others, such as UUJ170721_142, appear to exhibit seasonal ecotypes. We chose this MAG because this case could clearly highlight

the existence of seasonality of microdiversity dynamics. Some other viruses have a random type as we described “Additionally, some giant viruses displayed a high degree of variation, and many of the populations from different samples had alleles that differed from the ones in the reference MAG.” (LN252-254)

In response to your suggestion, we have generated an array of networks for the remaining eight persistent MAGs, now included as Fig. S11. This figure illustrates the presence of “ecotypes” within some viral populations. Apart from UJ170721_142 that has already been given in the main text, some other MAGs also display some degrees of seasonality more or less. For instance, UJ1705101_293 seems to exhibit a distinct spring ecotype across two years. And UJ180219_193 shows a separation between winter-spring and summer-fall groups.

Re: mSystems01168-24R1 (Genome-resolved year-round dynamics reveal a broad range of giant virus microdiversity)

Dear Prof. Hiroyuki Ogata:

Your manuscript has been accepted, and I am forwarding it to the ASM production staff for publication. Your paper will first be checked to make sure all elements meet the technical requirements. ASM staff will contact you if anything needs to be revised before copyediting and production can begin. Otherwise, you will be notified when your proofs are ready to be viewed.

Sincerely,
Xiyang Dong
Editor
mSystems

Reviewer #1 (Comments for the Author):

The authors have resolved all the comments I raised. Thank you all. I have no more comments.

Reviewer #2 (Comments for the Author):

Thank you to the authors for addressing my comments and providing a thorough revision. I have no further comments at this time, and I believe this manuscript will be an important contribution.